# Distributed coding of duration in rodent prefrontal cortex during time reproduction

Josephine Henke[1,2], Raven Bunk[1], Dina von Werder[1†], Stefan Häusler[1,2], Virginia L Flanagin[2,3], Kay Thurley[1,2]*

[1]Faculty of Biology, Ludwig-Maximilians-Universität München, Munich, Germany; [2]Bernstein Center for Computational Neuroscience Munich, Munich, Germany; [3]German Center for Vertigo and Balance Disorders, Ludwig-Maximilians-Universität München, Munich, Germany

**Abstract** As we interact with the external world, we judge magnitudes from sensory information. The estimation of magnitudes has been characterized in primates, yet it is largely unexplored in nonprimate species. Here, we use time interval reproduction to study rodent behavior and its neural correlates in the context of magnitude estimation. We show that gerbils display primate-like magnitude estimation characteristics in time reproduction. Most prominently their behavioral responses show a systematic overestimation of small stimuli and an underestimation of large stimuli, often referred to as regression effect. We investigated the underlying neural mechanisms by recording from medial prefrontal cortex and show that the majority of neurons respond either during the measurement or the reproduction of a time interval. Cells that are active during both phases display distinct response patterns. We categorize the neural responses into multiple types and demonstrate that only populations with mixed responses can encode the bias of the regression effect. These results help unveil the organizing neural principles of time reproduction and perhaps magnitude estimation in general.

**\*For correspondence:**
thurley@bio.lmu.de

**Present address:** †Institute of Medical Technology, Brandenburg University of Technology Cottbus-Senftenberg, Cottbus, Germany

**Competing interest:** The authors declare that no competing interests exist.

## Editor's evaluation

This study investigates the neural underpinnings of the bias property of timing, namely an overestimation for short and underestimation for long intervals, during an interval reproduction task in the medial prefrontal cortex of gerbils. The key novel result is that only neural populations with mixed responses, including ramping activity with linear increasing and slope-changing modulations as a function of reproduced durations, can encode the bias effect. Overall, experiments and data analysis are technically sound, and the conclusions well supported.

## Introduction

Animals including humans estimate the magnitude of physical stimuli, integrate path length, and keep track of duration to gather behaviorally relevant information from their environment. Although such estimates may ultimately be used for binary actions, like discriminating items or events and making decisions, the estimation itself is done on a continuum of values. Behavioral analyses over the past century established specific biases in magnitude estimation (e.g., reviewed in *Petzschner et al., 2015*) such as the *regression effect*, that is, the overestimation of small and the underestimation of large stimuli across a range of values (also known as regression to the mean, central tendency, or Vierordt's

law). Recently, this bias regained attention as it may be the result of an error minimization strategy (*Jazayeri and Shadlen, 2010*; *Petzschner and Glasauer, 2011*; *Cicchini et al., 2012*).

Despite a long history of behavioral research on magnitude estimation, its neural basis is not well understood. It is an ongoing debate whether a dedicated or distributed magnitude system exists in the brain (for review, see *Cohen Kadosh et al., 2008*; *Bueti and Walsh, 2009*; *Opstal and Verguts, 2013*). Human studies identified frontal, parietal, and striatal brain regions that are active during magnitude estimation. Recent studies with nonhuman primates used time interval reproduction experiments to investigate the connection between neural population dynamics in frontal and parietal cortices and magnitude estimation behavior (*Jazayeri and Shadlen, 2015*; *Sohn et al., 2019*). These studies focused on the estimation of time intervals lasting hundreds of milliseconds. What remains unclear is how their findings translate to durations of several seconds, that is, to time scales that are relevant for more complex and ecologically important behaviors like spatial navigation and action planning. Furthermore, it is unresolved to what extent the results generalize to nonprimate species.

We addressed these issues for Mongolian gerbils (*Meriones unguiculatus*) and designed a psychophysical task for time interval reproduction of several seconds on a continuous range. The task was implemented in virtual reality (*Thurley and Ayaz, 2017*), which allows for the precise control of the behaviorally relevant variables. We used gerbils because we could successfully train these animals in complex virtual reality tasks before (*Thurley et al., 2014*). We and others also performed timing experiments with gerbils in virtual reality (*Kautzky and Thurley, 2016*) or alike (*Shankar and Ellard, 2000*).

First, we demonstrate the capability of gerbils to precisely measure and reproduce time intervals of several seconds. We show that the gerbils' responses display the regression effect, indicative of an error minimization strategy. Then, we present associated neural activity in gerbil medial prefrontal cortex (mPFC) – a brain area that has been implicated in interval timing in rodents and other species (*Genovesio et al., 2006*; *Kim et al., 2013*; *Xu et al., 2014*; *Emmons et al., 2017*). The activity we observed was composed of mixtures of responses including phasic activation and ramp-like firing patterns, response types well known from the interval timing literature (e.g., *Mita et al., 2009*; *Merchant et al., 2011*; *Kim et al., 2013*; *Gouvêa et al., 2015*; *Paton and Buonomano, 2018*). Since our task involves measurement and reproduction, that is, the timing of an external event and of one's own behavior, we could test how individual cells participated in both. To make the variety of responses accessible, we provide a comprehensive characterization of activity at the single neuron level and show that, despite the response heterogeneity, the mPFC population jointly measures and reproduces time intervals lasting several seconds. We find that different types of ramping neurons are necessary to explain the regression effect and thus to gain a mechanistic understanding of the neural basis of time reproduction and perhaps of magnitude estimation in general. This reveals that variables underlying cognitive functions may be encoded by mixed responses within a local neural population.

## Results

### Behavioral characteristics of time reproduction in gerbils

We trained gerbils to measure and reproduce the duration of time intervals lasting a few seconds. After the presentation of a black screen, the animals had to reproduce its duration by walking along a corridor in virtual reality. Intervals were randomly sampled between 3 and 7.5 s. *Figure 1A–C* details the apparatus and task.

The gerbil's behavior exhibited typical magnitude estimation characteristics. Across the range of stimuli, small stimuli were overestimated whereas large stimuli were underestimated, that is, the regression effect (*Petzschner et al., 2015*). *Figure 1D* gives an example from a single experimental session. To quantify the regression effect across sessions and animals, we calculated the slope of linear fits between stimuli and reproductions in each session. The slopes increased during training (*Figure 1—figure supplement 2D*), indicating learning and improvement in the time reproduction task, but remained at levels less than 1 in the actual experimental sessions due to the regression effect (*Figure 1E*). Variability, that is, the coefficient of variation (CV), decreased with training and remained at low levels during the actual experiments (*Figure 1E*, *Figure 1—figure supplement 2D*). The CV correlated with the strength of regression, indicating a connection between the regression effect and

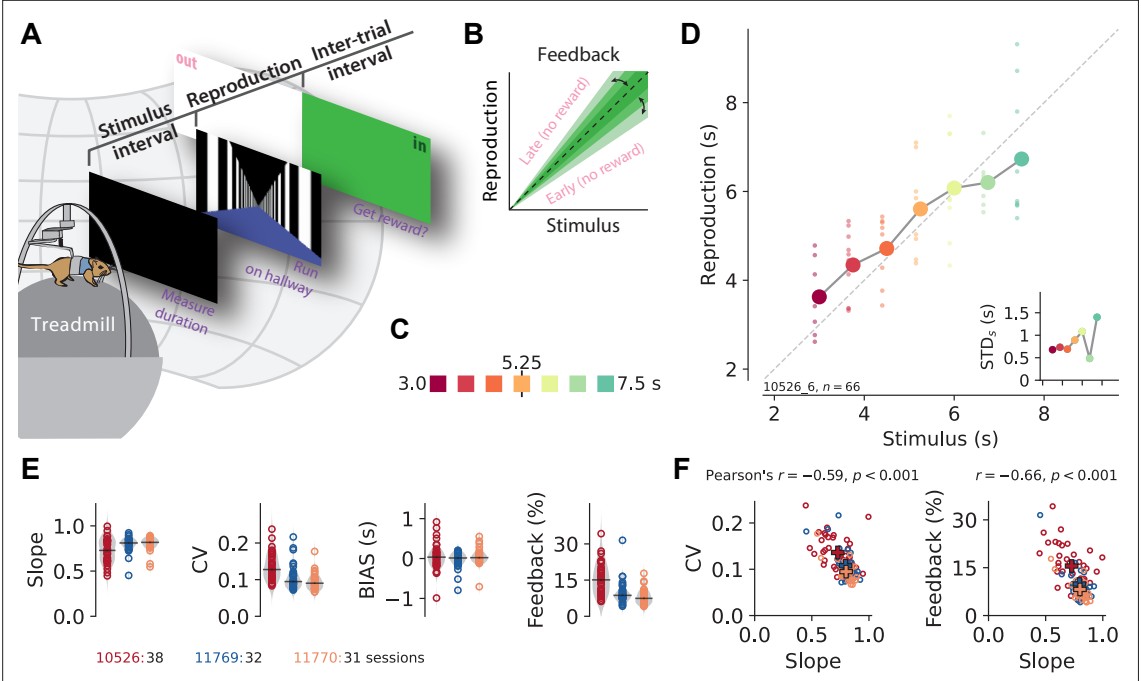

**Figure 1.** A time reproduction task for rodents. (**A**) Experimental apparatus and task. A gerbil was placed on top of a treadmill surrounded by a projection screen. Each trial started with a timed stimulus (black screen). The animal had to measure its duration and, when a virtual linear corridor appeared, reproduce the duration by walking. If the reproduction was close to the stimulus duration ('in'), a food reward was delivered and the entire screen was set to green for 3–4 s before another trial was initiated; otherwise, screen color was set to white ('out'). (**B**) The feedback range was narrowed/widened after each in/out response. (**C**) Stimulus intervals were randomly sampled from a discrete uniform distribution with seven values between 3 and 7.5 s. Colors identify stimulus duration and will be used throughout the paper. (**D**) Behavioral responses exhibited specific characteristics. Single reproductions (small dots) and their averages (large connected circles) showed the *regression effect. Inset:* standard deviation increased with stimulus duration (*scalar variability*). Same *x*-axis as in the main panel. Data in the main panel and inset are from one example session (of animal 10526). (**E**) Slope of the linear regression between stimuli and reproductions – quantifying the strength of the regression effect, with values closer to 1 meaning less regression, – coefficient of variation (CV), average bias, and average tolerance of the feedback range for each session sorted by animal. Values from single sessions are displayed as open circles. Gray violin plots illustrate distributions, and black solid lines mark the medians. Color identifies animals; see also ID numbers below the panels. (**F**) Slope negatively correlates with CV and feedback range across animals and sessions, indicating stronger regression effects with more variable responses. Open circles correspond to single sessions. Crosses mark averages for each animal. Color code as in (**E**).

The online version of this article includes the following source data and figure supplement(s) for figure 1:

**Source data 1.** Source data for *Figure 1E&F* and *Figure 1—figure supplement 2A–C*.

**Source data 2.** Source data for *Figure 1—figure supplement 2C₁ and C₂* and *Figure 1—figure supplement 3*.

**Source data 3.** Source data for *Figure 1—figure supplement 2D*.

**Figure supplement 1.** Behavioral data from an example session.

**Figure supplement 2.** Reward rates, reaction times, and training data.

**Figure supplement 3.** Speed vs. other behavioral parameters.

error minimization (*Figure 1F*). Some sessions showed a general under- or overestimation in addition to the regression effect (bias, *Figure 1E*).

To keep the animals motivated, we gave them feedback on their reproduction performance (*Figure 1A*). Although this feedback could be used to correct reproduction errors, we still observed the regression effect. Since we adapted the feedback range after every stimulus (*Figure 1B*), its width indicated estimation precision, with a narrower feedback range corresponding to higher precision. The average width of the feedback range in a session correlated negatively with the slope, which is another sign of a connection between the regression effect and error minimization (*Figure 1F*).

## Single cells differentially encode time during measurement and reproduction

We recorded a total of 1766 mPFC units over 101 experimental sessions from three gerbils. Visual inspection of the spiking responses, after sorting by stimulus duration and splitting into the two task phases 'measurement' and 'reproduction,' revealed a variety of response patterns that could underlie time reproduction. Three examples are shown in *Figure 2*; further examples can be found in *Figure 2—figure supplement 2-Figure 2—figure supplement 4*. The neuron in *Figure 2A* increased its activity during measurement, such that the firing rate by the end of the phase correlated with the stimulus interval (see *Figure 2D*). During reproduction, this neuron displayed downward ramping. However, here the change in firing rate correlated with the stimulus interval to be reproduced, such that for shorter intervals the firing rate decreased faster than for longer intervals (see also *Figure 2D*). This effect was also observed in cells whose activity increased to a fixed level at the end of reproduction that was independent of the stimulus duration (*Figure 2C and D*). Such ramp-to-threshold cells peaked at a constant time from the end of the reproduction epoch, with higher slopes for shorter intervals, suggesting the prediction of time to an event (*Merchant and Georgopoulos, 2006*; *Murakami et al., 2014*). Yet other cells constantly increased firing rate (*Figure 2B*), similar to what we saw during measurement in the neuron in displayed in *Figure 2A*. We also found cells that responded at absolute times (*Figure 2—figure supplement 3*) or relative to the reproduction interval, for example, at its begin or end (*Figure 2—figure supplement 4A*).

Between measurement and reproduction, the neurons adapted their response patterns. We did not find a single cell that responded in the very same way in both task phases and essentially repeated its activity pattern from the measurement phase during reproduction – an observation that will be analyzed systematically below.

Since reproduction involved walking in our behavioral task, we looked at dependencies between firing rate and running speed. In most example neurons, running speed influences were weak. Although speed response functions may have been significant (from shuffled controls, see Materials and methods), modulation of the speed response function was low (*Figure 2—figure supplement 2-Figure 2—figure supplement 4*). Still, some neurons displayed an obvious speed modulation. For instance, the shape of the spike density functions (SDFs) of the neurons in *Figure 2—figure supplement 3G and H* was rising and decreasing over the reproduction phase, and this pattern corresponded well to changes of the running speed. These neurons had a monotonously rising speed response function. Yet other neurons had large speed modulation indices, but the overlap between their SDFs and the running speed pattern was small (*Figure 2—figure supplement 3C–F*). Velocity changes over a trial certainly contributed to the response of such neurons but could not fully explain their firing. To assess speed modulation more systematically, we calculated modulation indices of all neurons for virtual speed and for running speed on the treadmill. These modulation indices were widely distributed and often larger for running speed on the treadmill than for virtual speed. Although also weak speed modulation could be significant, it was significant in only 22% of the neurons for virtual speed and in 27% for the running speed (*Figure 2—figure supplement 5*).

We note that running speed as well as other unobserved behavioral factors may contribute to firing in some neurons and hence may explain differences in firing patterns between task phases. However, since we were interested in the collective action of the neuronal population, we did not investigate this further and also decided against excluding such neurons from the subsequent analyses.

## Single-cell picture holds for the whole population

The different response patterns for single neurons were also obvious when visualizing the whole population. Some neurons ramped up at the end of measurement, others were active at the beginning and then decreased activity (*Figure 3A*). During reproduction, a similar but more pronounced pattern emerged with up/down ramping and phasically active neurons (*Figure 3B*).

Striking, however, were the activity differences between measurement and reproduction, indicating a state change in the population between both task phases. When individual cells were sorted in the same order for both measurement and reproduction, no global pattern was visible (*Figure 3B and C*). Also, correlating population vectors in corresponding time bins for measurement and reproduction yielded only low values (*Figure 3D*). For single cells, however, the correlation between task phases was larger and significant in about 20% of the cells (*Figure 3E*). Note that this does not indicate a

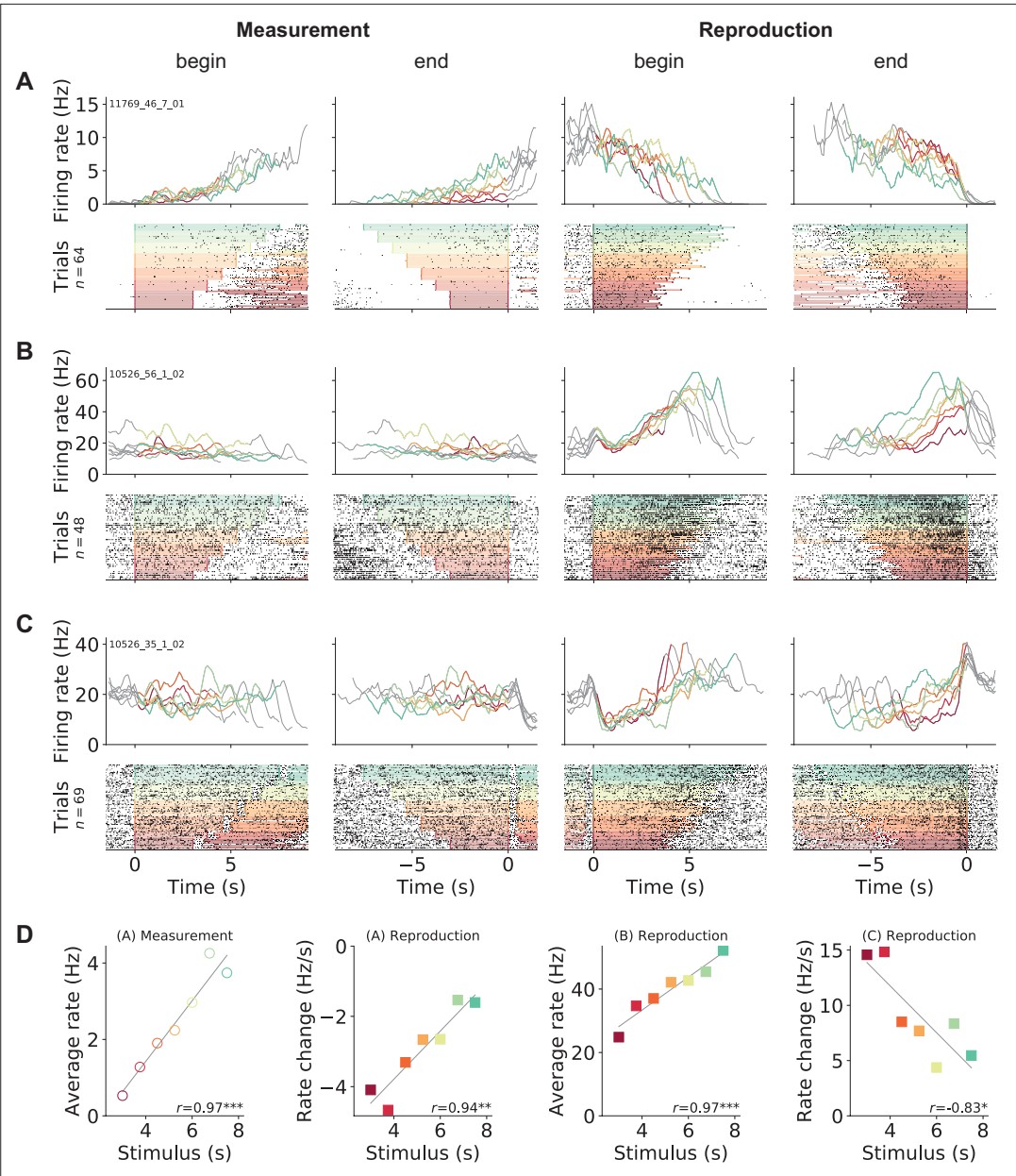

**Figure 2.** Gerbil medial prefrontal cortex (mPFC) neuron responses during time reproduction. (**A**) A cell that linearly increased its firing rate during measurement and ramped down to zero during reproduction. (**B**) A neuron that scaled its firing with the stimulus duration during reproduction and (**C**) a ramp-to-threshold cell. (**A–C**) Panels display spike raster plots sorted by stimulus duration (bottom) and corresponding spike density functions (SDF, top). Each column plots the data with different alignment, that is, measurement begin and end, reproduction begin and end. Color identifies stimulus duration as in *Figure 1C*. In the raster plots, black ticks are single spikes. For better visualization, we only plot half of the spikes (randomly chosen). Measurement or reproduction phases are delimited by underlayed color. The SDFs are colored in the respective task phase, outside they are displayed as thin gray lines. (**D**) Dependence of firing on stimulus duration in the example cells. Single markers show the average firing rate or change of firing rate at each stimulus duration. Open dots are used for data from measurement and filled squares for those from reproduction. Solid lines are linear fits. Pearson's correlation coefficient and significance is given in the lower-right corner. Above each panel, cell and task phase is indicated. The averages of firing rate and its change were calculated from the last half of the SDFs in the corresponding task phase.

The online version of this article includes the following figure supplement(s) for figure 2:

*Figure 2 continued on next page*

*Figure 2 continued*

**Figure supplement 1.** Electrophysiological recordings in gerbil medial prefrontal cortex (mPFC) – histology and spike sorting.

**Figure supplement 2.** Ramping neurons.

**Figure supplement 3.** Phasically timing neurons.

**Figure supplement 4.** Other example neurons.

**Figure supplement 5.** Influence of speed on single neuron responses.

precise correspondence between the activity profiles for measurement and reproduction but rather that a neuron was active in both phases. See, for example, the neuron in *Figure 2A*, which shows a negative correlation between task phases.

Neural activity was similar for different stimulus durations during reproduction. On the one hand, population activity correlated for different stimuli (*Figure 3F₂*); on the other hand, single-neuron activity correlated across different stimuli in about 20% of the neurons (*Figure 3G₂*). During measurement, correlations across stimuli were absent (*Figure 3F₁ and G₁*).

The above picture remained, when we split activity into odd and even trials. Activity was largely similar arguing for stable neural activity throughout a session. However, during measurement, data was more noisy in agreement with the weak correlations we observed in this task phase (*Figure 3— figure supplement 1*).

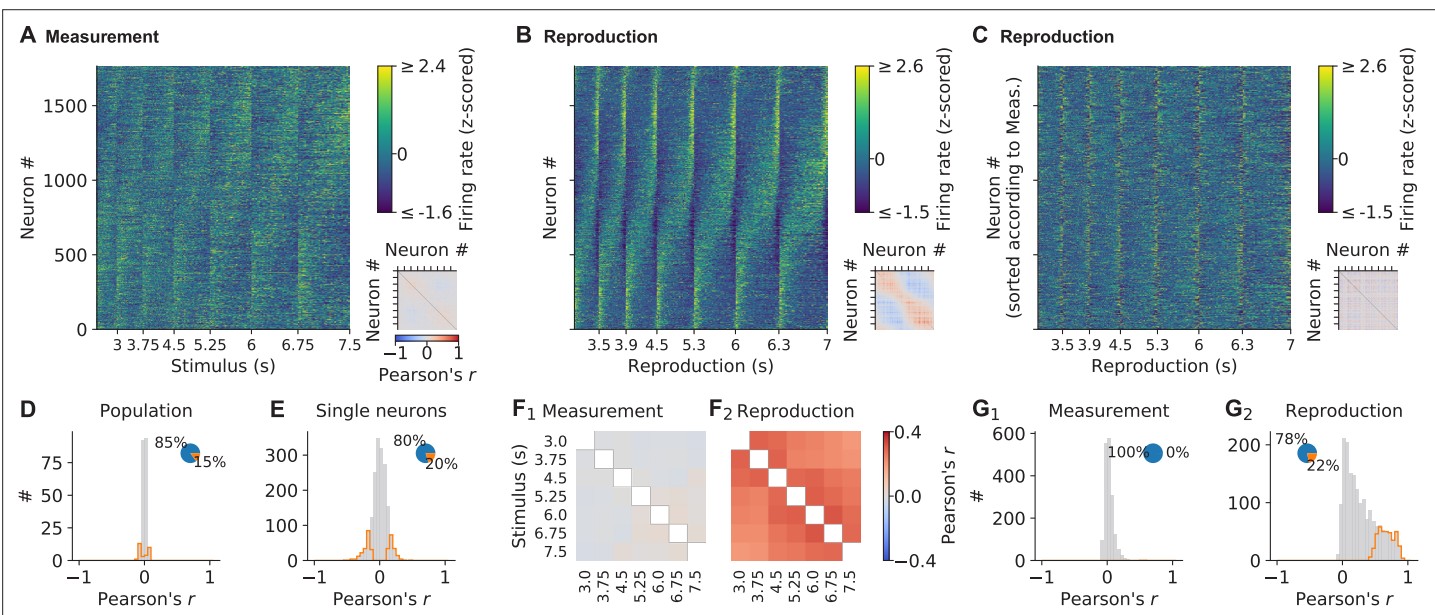

**Figure 3.** Single neuron and population dynamics in measurement and reproduction. (**A**) Normalized (z-scored) spike density functions (SDFs) of all 1766 medial prefrontal cortex (mPFC) neurons for each stimulus interval during the measurement phase sorted by their timing within the intervals. *Small panel:* matrix of pairwise Pearson correlations between all neurons. Diagonal entries not plotted. (**B**) Same as (**A**) but for the reproduction phase. (**C**) Same as (**B**) but with neurons sorted as in (**A**). (**D**) Population vector correlations in corresponding time bins of measurement and reproduction. (**E**) Pearson correlations between measurement and reproduction for single neurons. (**F**) Pairwise correlations of the whole population for different stimuli in measurement (**F₁**) and reproduction (**F₂**). Diagonal entries not plotted. (**G**) Distributions of average Pearson correlations of single-cell activity for different stimuli in measurement (**G₁**) and reproduction (**G₂**). Histograms in (**D, E, G**) are displayed in gray with significant values ($p<0.05$) delimited by an orange outline. Pie plots show significant (orange) and nonsignificant (blue) percentages.

The online version of this article includes the following source data and figure supplement(s) for figure 3:

**Source data 1.** Spike density function (SDF) source data for *Figure 3A and B* and all upcoming analyses and figures based on SDFs.

**Figure supplement 1.** Stability of neuronal responses.

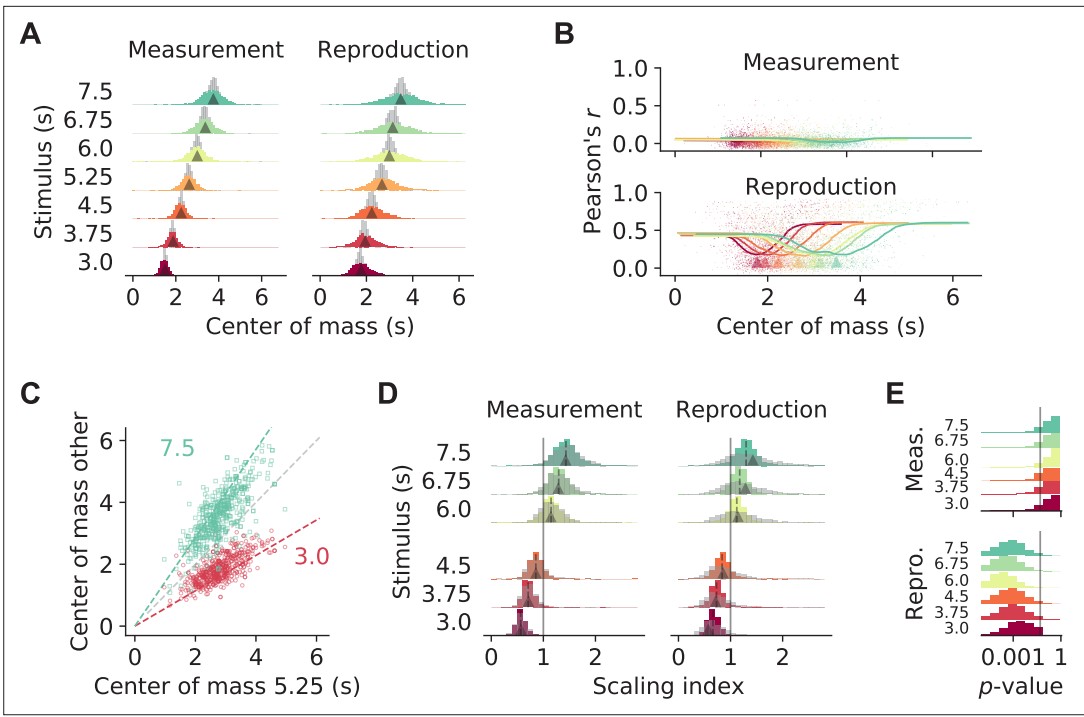

**Figure 4.** Temporal scaling. (**A**) Distributions of center of mass for all 1766 medial prefrontal cortex (mPFC) neurons (colored histograms). Gray histograms give distributions of center of mass for noise spike density functions (SDFs). Arrowheads mark the middle of the stimulus interval and of the average reproduced interval, respectively. (**B**) During reproduction, the average Pearson correlations of single-cell activity for different stimuli (*Figure 3F*) were larger for neurons with center of mass at the begin or end of an interval. Dots give single-cell data, solid lines are moving averages. Arrowheads in the lower panel mark the middle of the average reproduced interval. (**C**) Center of mass of each cell for 5.25 s against 3 s (red) and 7.5 s (green) during reproduction. Colored dashed lines mark the predicted ratio; gray dashed line corresponds to no change. (**D**) Distributions of scaling indices for all cells, that is, center of mass of the SDF at every stimulus divided by the center of mass for 5.25 s. Solid vertical line marks 1. Dashed lines mark the medians and arrow heads the ratio between 5.25 s and the respective other stimulus duration. Gray histograms give distributions of center of mass for shuffled control SDFs. (**E**) Bootstrapped distributions of p-values for Kolmogorov–Smirnov tests between scaling indices for recorded and shuffled SDFs. Solid vertical line marks 5%. Logarithmic x-axis.

## Scaling of neuronal activity with duration

The activity of the prefrontal neurons we recorded appeared to scale with stimulus duration (*Figure 3A and B*), which was reminiscent of findings in interval timing studies for striatum (*Mello et al., 2015*) and prefrontal cortex (*Xu et al., 2014*; *Wang et al., 2018*).

To examine stimulus-related scaling in our data set, we followed the approach of *Mello et al., 2015*. We first calculated for each cell the center of mass (COM) of the SDF at every stimulus duration. The COMs were, in particular during reproduction, widely distributed and tiled the time intervals. For comparison, we simulated SDFs with no temporal modulation ('noise'). The corresponding COMs clustered around the center of the intervals and significantly differed from those for the recorded SDFs ($p<0.001$ in all cases, Kolmogorov–Smirnov test; *Figure 4A*). Interestingly, during reproduction, for neurons with a COM at the begin or end of an interval, activity correlated more strongly across different stimuli (*Figure 4B*, *Figure 3G₂*). This indicates that, in particular for neurons with increased firing at the border of the interval, activity depended on the temporal stimulus.

As a second step, we calculated scaling indices by dividing every cell's COMs by the corresponding COM at the 5.25 s stimulus, that is, the mean of the stimulus distribution. These scaling indices theoretically range between $3.0/5.25 \approx 0.57$ and $7.5/5.25 \approx 1.43$. Indeed, scaling indices were close to the theoretical values during both measurement and reproduction (*Figure 4C and D*). During reproduction, scaling indices were even larger or smaller than the prediction in a manner consistent with the regression effect, that is, smaller durations had larger scaling factors and vice versa.

Finally, we obtained scaling indices after shuffling cell identities across stimuli to test whether the scaling indices from the recorded data were in fact the result of a meaningful modulation. Only for the reproduction phase the indices from shuffling were more widely distributed than the data; for the measurement phase, a difference was not clearly visible (*Figure 4D*). Statistical testing, however, identified significant effects in both phases ($p < 0.01$ in all cases, Kolmogorov–Smirnov test). This statistical result was likely due to the large number of samples contained in the data set. Therefore, we performed bootstrapped Kolmogorov–Smirnov tests on 10% of the data over 10,000 runs. The p-values obtained lay below 0.05 for almost all cases during reproduction. During measurement, only about 30% of the bootstrapped Kolmogorov–Smirnov tests were significant (*Figure 4E*).

Taken together, this indicates activity-dependent temporal scaling of prefrontal single-cell responses in the reproduction phase alluding to a potential role in time encoding and the regression effect. Scaling during measurement was not different from what would be expected of neural activity that unfolds over time. Note that the results are in line with the non-/significant correlations of activity across stimuli for measurement and reproduction, respectively (*Figure 3F and G*).

## Collective population activity in the different task phases

The activity differences between measurement and reproduction revealed above argue against an underlying mechanism just at the single-cell level. Therefore, we examined the collective properties of the prefrontal neurons by decomposing population activity into its principal components (PCs). We used demixed principal component analysis (PCA), a form of PCA that allowed us to separate time course-related contributions to neural activity from contributions that depended on stimulus duration (*Kobak et al., 2016*).

Interestingly, the PCs for measurement and reproduction shared common features (*Figure 5A and B*). The time course-related PCs were ramp-like (PC 1) or had a comparable non-monotonous shape (PC 2 and 3). The strongest stimulus-dependent PC was constant over time and had amplitudes that were ordered by stimulus (see bottom-right panels in *Figure 5A and B*). Moreover, the contribution of the PCs to the single-cell responses (PCA scores) correlated for PC 1 and stimulus PC 1, indicating a link between ramp-like and stimulus-dependent activity in both measurement and reproduction (*Figure 5C*, see also *Figure 5—figure supplement 4*). Nevertheless, differences between measurement and reproduction were also obvious in the collective activity. In the measurement phase, PCs 2 and 3 in general contributed very little to explaining population activity; instead, PCs were most strongly driven by stimulus duration (*Figure 5—figure supplement 1*). During reproduction but not during measurement, stimulus PC 1 also correlated with PCs 2 and 3, arguing for a richer representation of the to-be-reproduced compared to the to-be-measured time interval.

Recent work has shown that collective neural activity evolves along very similar trajectories when estimating different time intervals, but which final state is reached depends on the duration of the interval. In contrast, when a time interval is reproduced, trajectories of similar length are found for different stimuli but the speed of progression decreases with duration (*Gouvêa et al., 2015*; *Wang et al., 2018*; *Remington et al., 2018*; *Sohn et al., 2019*). Since we separated time course-related from stimulus information through the demixed PCA, we could not see such stimulus-dependent effects directly. We therefore also applied conventional PCA to our data. The strongest PCs from conventional PCA were similar in shape to those from demixed PCA but were – as expected – influenced by the stimulus (*Figure 5D and E*, *Figure 5—figure supplement 5*). Indeed, in the measurement phase, neural activity developed at similar speed along trajectories with a length that depended on stimulus duration. During reproduction, trajectories had similar length. However, activity along those trajectories accelerated over an interval resulting at higher average speed for stimuli of shorter duration. Note that this effect corresponds to the temporal scaling described in the previous section.

It is possible that the population responses during measurement and reproduction are not actually collective phenomena but simply the result of pooling single neurons. To test this, we compared population activity to surrogate data that preserve stimulus tuning of single neurons, correlations of single-cell firing rates across time, and pairwise correlations between neurons (*Elsayed and Cunningham, 2017a*). During measurement, PC 1 of the original data was larger than expected, suggesting that collective activity added to representing ongoing time; similarly, PC 3 was larger than expected during reproduction (*Figure 5—figure supplement 6*). Stimulus PC 1 was stronger than expected in

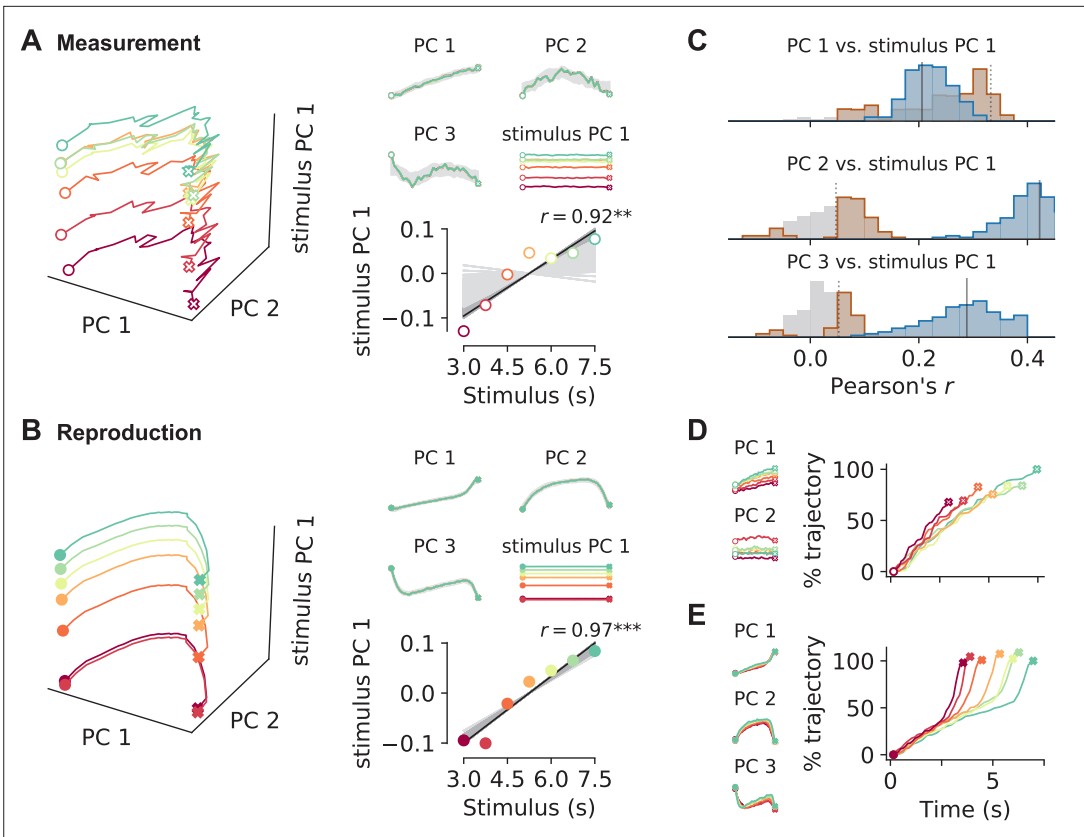

**Figure 5.** Decomposition of population activity. Demixed principal component analysis (PCA) yields separate principal components (PCs) for the time course during a trial (PCs 1–3) and for each stimulus duration (stimulus PC 1). (**A**) PCs for measurement and (**B**) for reproduction. The small panels display individual PCs at each stimulus interval scaled to the length of the interval. Note that PCs 1–3 for different stimuli lie on top of each other, demonstrating perfect demixing. Stimuli are colored as in previous figures. Circles and crosses mark interval start and end. Open symbols are used for measurement and filled for reproduction. The PCs were calculated for the whole data set and gray-shaded areas delimit the standard deviation of results from bootstrapping with 10% of the cells. Bottom-right panels in (**A**) and (**B**): correlation of stimulus PC 1 with the stimulus duration. Black solid line is a linear fit, and gray lines are fits for the bootstrap samples; dark gray significant, light gray nonsignificant cases. (**C**) Distributions of Pearson's correlation coefficients between PCs 1–3 and stimulus PC 1 from bootstrap samples. Significant correlations are colored in brown (measurement) and blue (reproduction). Dotted (measurement) and solid (reproduction) black lines mark coefficients for the whole data set. (**D**, small panels) PCs for conventional PCA during measurement scaled to the length of the interval for each stimulus. Note that, in contrast to demixed PCA, conventional PCs contain both time course-related and stimulus information. (**D**, large panel) Time course of PC 1 for conventional PCA at each stimulus during measurement. The percentage of covered trajectory is given with respect to PC 1 for the 7.5 s stimulus. (**E**) Same as (**D**) for reproduction.

The online version of this article includes the following figure supplement(s) for figure 5:

**Figure supplement 1.** Explained variance of demixed principal component analysis (PCA).

**Figure supplement 2.** Proper demixing of principal components (PCs) by demixed principal component analysis (PCA).

**Figure supplement 3.** Demixed principal component analysis (PCA) results are similar across animals.

**Figure supplement 4.** Distributions of and correlations between demixed principal component analysis (PCA) scores.

**Figure supplement 5.** Decomposition with conventional principal component (PC) analysis.

**Figure supplement 6.** Collective activity adds to principal components (PCs).

reproduction, indicating that here the population as a whole contributed to the representation of the stimulus interval.

## Heterogeneous prefrontal activity can be categorized into a few response types

To extract response types from a set of neurons, usually, specific tests must be designed, which are based on predefined knowledge of the responses of interest. To circumvent such rigid preselection, we used the demixed PCA results for categorizing neurons into different response types.

Single-cell activity patterns can be decomposed into linear mixtures of different PCs. We focused on the contributions of the strongest PCs, which were time course-related PC 1 and stimulus PC 1 during measurement and PCs 1–3 and stimulus PC 1 during reproduction (*Figure 5—figure supplement 1*). As we already mentioned above, the PCA scores for single-cell responses were correlated for these PCs (*Figure 5C*), pointing to a potential utility for capturing single-cell responses. For instance, mixing the ramp-like PC 1 and stimulus PC 1 with its constant responses ordered by stimulus, one can

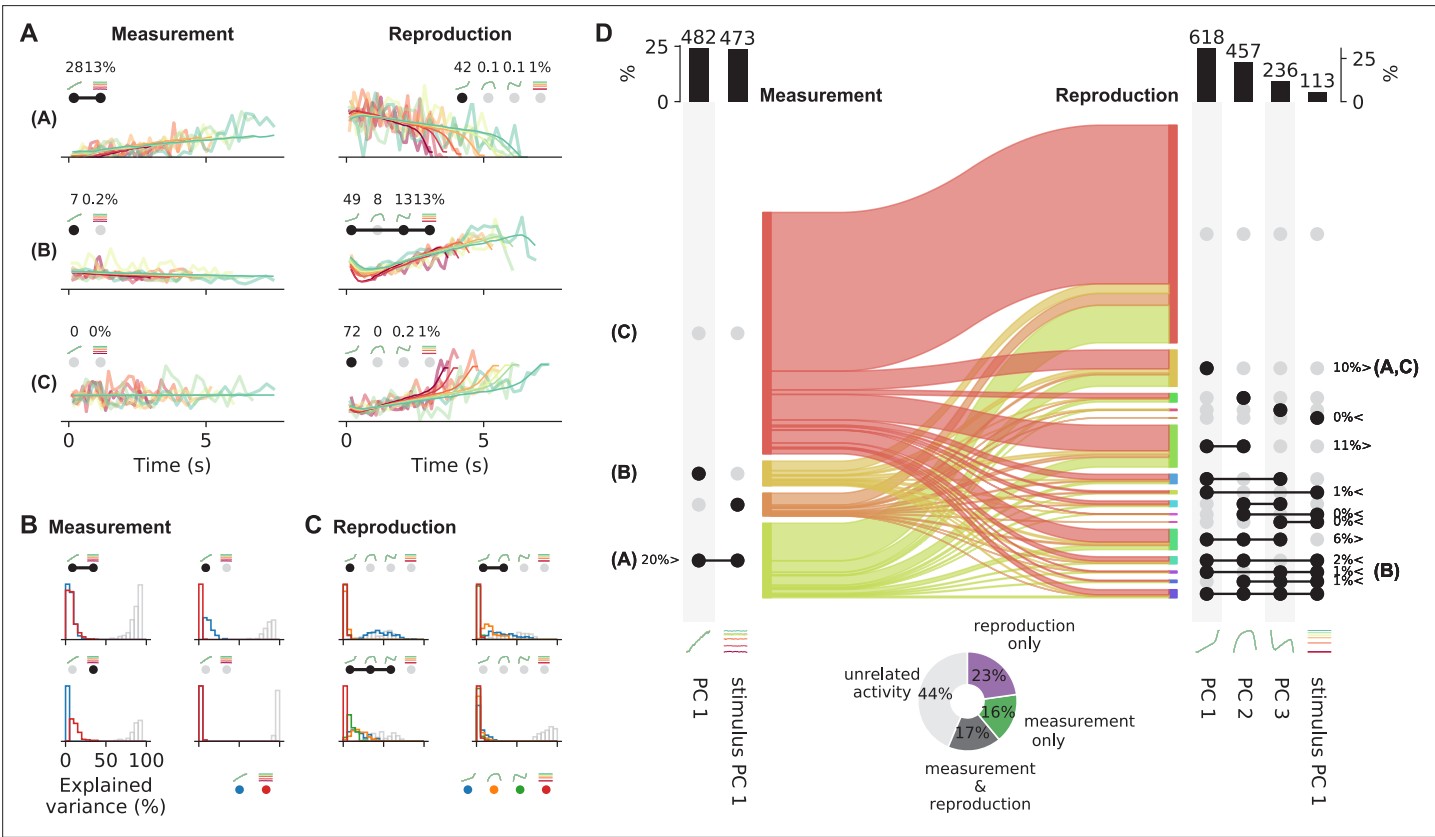

**Figure 6.** Single-cell activity and their contribution to the population response change from measurement to reproduction. (**A**) Reconstructions (thin lines) of the firing patterns (spike density functions [SDFs], thick faint lines) of the example neurons from *Figure 2* (identified by bold letters; note that here SDFs were not smoothed). Markers show percent explained variance for each principal component (PC) and illustrate the response type according to the categorization (see **D**). For instance, the neuron in the first row is reconstructed by the linear combination of PC 1 and stimulus PC 1 during measurement and with (negative) PC 1 during reproduction. (**B**) Distributions of variance of single-cell responses explained by PCs during measurement. Variance explained by PC 1 is displayed in blue and by stimulus PC 1 in red. The remainder (i.e., variance unexplained) is displayed as gray open bars. Each panel displays distributions for cells belonging to one of the four possible response categories visualized by the markers above the panel. (**C**) Same as (**B**) for the reproduction phase. Here the coloring is PCs 1–3, blue, orange, green; stimulus PC 1, red. Distributions are displayed for the four categories with the most cells. (**D**) Transition between response categories from measurement to reproduction (Sankey diagram). At the margins the numbers of cells are given with activity patterns that can be reconstructed by PCs. For measurement (left), PC 1 and stimulus PC 1 were considered and for reproduction (right) PCs 1–3 and stimulus PC 1. Dark dots indicate the contribution of a PC to a response category. Percentages are only given for categories that contain significant numbers of cells compared to shuffled data. The signs < and > indicate if the number is smaller or larger than for shuffled data. Zero percentages mean less than 1%. Bold letters correspond to the examples from (**A**). Bar graphs at the top show percentages and cell numbers with contributions of each PC. Pie chart at bottom shows percentages of cells active during the task phases determined from the reconstructions.

describe the activity during the measurement phase of cells like in *Figure 2A*. Similarly, mixtures of the different time-modulated profiles from PCs 1–3 and stimulus PC 1 can capture various response patterns during reproduction (*Figure 6A*).

Combinations of PC 1 and stimulus PC 1 yield three possible response categories in the measurement phase: activity explained by (1) PC 1 only, (2) stimulus PC 1 only, or (3) both PCs. For each neuron, a category was selected based on the angle between its PC 1 and stimulus PC 1 scores (*Figure 5—figure supplement 4C*). We only included cells with large demixed PCA scores (explained variance) in this analysis (see also *Figure 5—figure supplement 4*). Cells with small scores were categorized as 'unrelated activity,' giving a fourth response category. Likewise, 15 different categories were defined for the reproduction phase from combinations of PCs 1–3 and stimulus PC 1 plus a 16th category for 'unrelated activity'.

Categorization provided meaningful representations of single-cell responses as reflected in the variance explained by the contributing PCs (*Figure 6B and C*). For instance, large parts of variance were explained by PC 1 for neurons in the category 'PC 1 only' during measurement; contributions from stimulus PC 1 were marginal. In the stimulus PC 1-only category, the situation was reversed, and for the mixed PC 1 and stimulus PC 1 category contributions by both PCs matched (*Figure 6B*). However, only less than 50% of variance could be explained by the two PCs in the measurement phase. The PCs used for reconstructing during the reproduction phase better captured the activity; leaving less variance unexplained (*Figure 6C*). In particular, for the PC 1-only category often cell activity could be explained to more than 50% by PC 1.

Cells that could be described by PC 1 had a ramp-like response profile. During measurement, about a quarter of the cells showed such ramping activity. These cells included ramping to stimulus duration-dependent levels like the neuron in *Figure 2A* and ramping that did not depend on stimulus. The other cells represented the stimulus duration in a different way or showed activity unrelated to the task during measurement (67%; *Figure 6D*).

During reproduction, about 10% of the neurons showed ramping activity (PC 1 only), for example, ramp-to-threshold cells where the rate of ramping decreased with stimulus duration (examples in *Figure 2B and C*, *Figure 2—figure supplement 2B and C*). Another ~10% of neurons were explained by mixing PCs 1 and 2, and 6% by a mixture of PCs 1–3. Only 1% was explained by ramps that encoded stimulus (i.e., PC 1 and stimulus PC 1). However, counting all combinations of stimulus PC 1 with PCs 1–3 showed that about 5% of the neurons combined ramps with stimulus duration-dependent firing. In total, 35% of the cells contained a ramping component. These cells overlapped largely with those whose activity was significantly correlated across stimuli (not shown, *Figure 3G_2*). In *Figure 2—figure supplement 2*-*Figure 2—figure supplement 4*, further examples for the different categories can be found.

We also determined if the ramp-like responses (cells explained by PC 1 and those explained by PC 1 + stimulus PC 1) could be driven by running speed rather than time reproduction. Half of the ramping neurons were significantly modulated by virtual speed and 60% by running speed. Nevertheless, speed modulation indices were in general low for those ramping cells (*Figure 2—figure supplement 5*).

The categorization analysis strengthened the picture of changing response patterns between measurement and reproduction we have already noted above in several places. Cells switched response types between task phases but with no specific pattern as becomes obvious from the flow diagram in *Figure 6D*. Separating cells with small scores from those with large ones, we could estimate the number of cells active during the task phases. About 17% of the cells fired action potentials in both phases, 16% were only active during measurement and 23% only during reproduction; 44% of the cells did not contribute substantially in either phase (pie chart in *Figure 6D*).

## Time encoding is governed by different response types

To understand how mPFC may encode time in our task, we read out ongoing time from the responses of all neurons using a simple linear regression decoder (see Materials and methods). During the reproduction phase, time readouts were accurate in the first seconds; however, by the end of the phase, a regression effect was visible with over- and underestimation at short and long stimuli, respectively (*Figure 7A*). This was due to neurons with large PCA scores, as we obtained a similar result when decoding only from such neurons (*Figure 7—figure supplement 1*). In contrast, time readouts

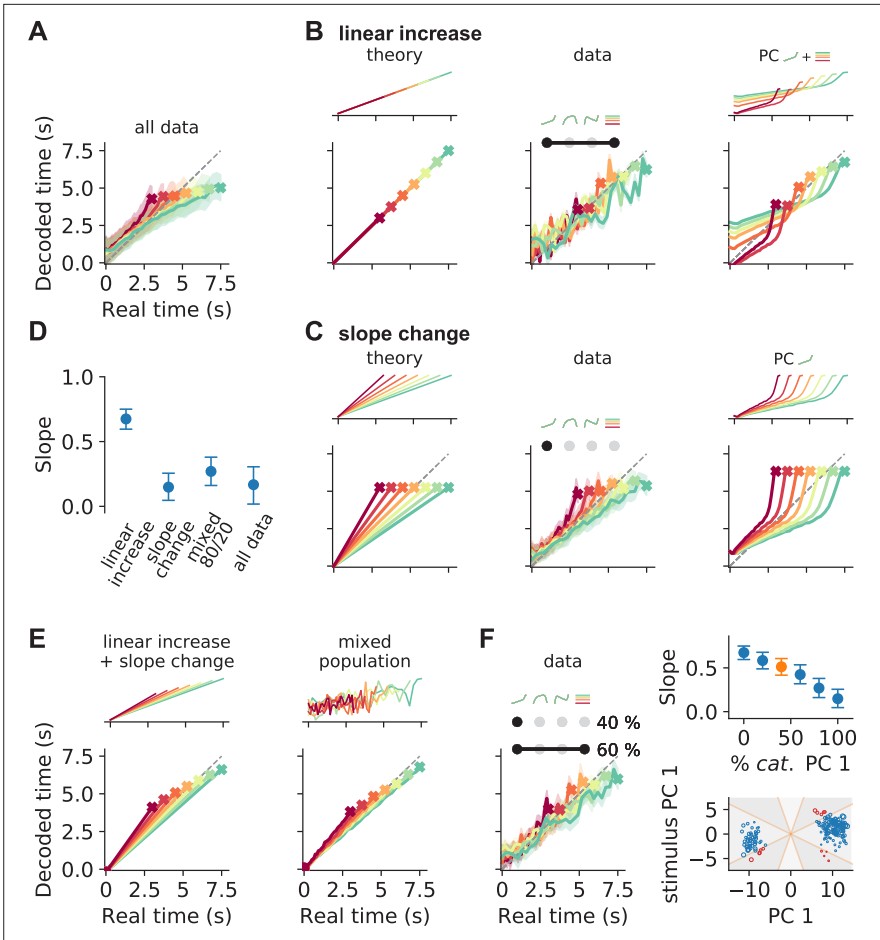

**Figure 7.** Decoding elapsed time from prefrontal population activity. (**A**) Elapsed time (average ± standard deviation from bootstrapping) decoded from the responses of all neurons recorded during the reproduction phase. As before, color identifies stimulus duration. Crosses mark final values. A strong regression effect is visible. (**B**) Time decoding with neurons that ramp to stimulus-dependent levels but with the same slope and (**C**) with ramp-to-threshold cells. Left panels illustrate the theoretical prediction for decoded time and example neuronal activity above. Middle panels plot decoding results using only neurons from the respective response category. Right panels give results for decoding using the corresponding components from demixed principal component analysis (PCA). As displayed in the top panels, these components are principal component (PC) 1 and stimulus PC 1 in (**B**); and PC 1 only in (**C**). (**D**) Median slopes of the linear regression between the final values of real and predicted time for decoding from data in (**A–C**) and for a mixture of 80% slope-changing and 20% linear increasing cells, which aligns well with decoding from all data (**A**); regarding mixtures see also (**F**). Error bars delimit interquartile ranges (from bootstrapping). (**E**) Mixtures of linear increasing activity and slope changes explain behavioral regression effects. Theoretical predictions for mixing both responses in single neurons (left) or as two different response types across a population (right). For the second case, a neuron with noisy linear increasing activity is displayed as an example. (**F**) Decoding results for a mixed population of 40% slope-changing and 60% linear increasing cells. The cells were sampled at these fractions in each bootstrapping run from the response categories we identified in our recorded data. The upper-right panel shows the regression slope (**D**) for different fractions of slope-changing cells. The orange marker corresponds to the example in the left panel. The lower-right panel displays the PCA scores of the cells from the linear increasing (red; PC 1 + stimulus PC 1) and slope-changing categories (blue; PC 1 only). The size of the marker illustrates the decoder weight $\beta$ for that cell. See also *Figure 5—figure supplement 4C*.

The online version of this article includes the following figure supplement(s) for figure 7:

**Figure supplement 1.** Decoding elapsed time from data, shuffled data, and noise.

**Figure supplement 2.** Decoding time from phasically active neurons.

**Figure supplement 3.** Decoding time during measurement.

using neurons with small PCA scores were very imprecise. Here, decoded time was almost constant throughout the reproduction phase, such that real time was overestimated initially and underestimated at the end of reproduction. A similar picture emerged when we read out time from shuffled data and was even more pronounced for pure noise (*Figure 7—figure supplement 1*).

The regression effect we observed when decoding from all neurons was stronger than in the behavior. The slope of the linear regression between the final values of real and predicted time was close to zero *Figure 7D*, values from behavior were between 1 and 0.5 (see *Figure 1E*). This discrepancy probably comes from the fact that the neuronal population includes not only neurons encoding the regression effect. We therefore wondered what response types could mediate the regression effect and simulated a few stereotypical cases. Decoding time from neurons that ramp with same slope to stimulus duration-dependent levels (linear increasing neurons) shows precise time representation but no regression effect (*Figure 7B*). Whereas ramp-to-threshold cells that change slope but reach same activity levels by the end of an interval can only encode the mean of the stimulus distribution and result in maximal regression (*Figure 7C*).

Decoding time from the response categories (*Figure 6*) corresponding to the theoretical cases led to similar results. For the response type combining time-dependent PC 1 with stimulus PC 1, time reproduction displayed only a weak regression effect (*Figure 7B*) and ramp-to-threshold responses (PC 1 only) displayed very strong regression (*Figure 7C*); see also *Figure 7D*. Decoding time directly from the demixed PCs gave similar results (*Figure 7C and D*).

Regression effects in our behavioral data lay between the extremes yielded by the two response categories (*Figure 1E*). This discrepancy made us think about another solution to the question of how regression effects may arise. In theory, combining ramping to stimulus duration-dependent levels with slope changes by stimulus also generates regression effects (*Figure 7E*). Such a combination can be implemented either with mixed response patterns within one neuron or as a mixture of response types across a neuronal population. To find out which of the two scenarios underlies our data, we looked at the distribution of PCA scores for the cells in the linear increasing and the slope-changing categories. These scores were broadly distributed and also the corresponding decoding weights did not reveal a particular structure, indicating time coding comprising different response types (bottom-right panel in *Figure 7F*). To further test this possibility, we mixed responses of recorded cells from both categories at different fractions. Decoding not only yielded regression effects, but the strength of regression could be manipulated by the relative shares of both response types. The more slope-changing cells were present in the population, the stronger was the regression effect (top-right panel in *Figure 7F*).

Another response type that has been often connected to timing are phasically active cells. We therefore wondered how ongoing time would be encoded by such neurons. Constructing again theoretical stereotypes, we saw that neurons that are active relative to the stimulus interval can only encode a single time point. In contrast, a population of neurons that are phasically active at different absolute times tiling the whole interval would provide an accurate time representation (*Figure 7—figure supplement 2*). Since neither relative nor absolute timing neurons by themselves predict regression effects as found in the behavioral responses, again a mixture of both response types would be required. Matching these theoretical results to our own recordings turned out to be less straightforward. We recorded phasically active cells (*Figure 2—figure supplement 3*), but this subset of cells did not completely tile the whole time intervals. However, such a prerequisite is necessary to seriously attempt to reproduce the theoretical predictions.

Finally, we examined whether time could also be decoded from the neuronal responses in the measurement phase. Here, decoding was imprecise with overestimation at the begin and underestimation at the end of the interval (*Figure 7—figure supplement 3*) – a picture that also appeared for shuffled and noisy data (*Figure 7—figure supplement 1*) and matches the less pronounced time signaling across the population during measurement, which we have found above in several places. A general underestimation was also found when we used only neurons in the PC 1 + stimulus PC 1 category, indicating their foremost influence on the collective readout. Interestingly, when we decoded time from the cells in the PC 1-only category, a regression effect was also seen during measurement (rightmost panel in *Figure 7—figure supplement 3*), suggesting an impact of previous stimuli (prior knowledge) on the activity of these neurons already during stimulus measurement and not just during reproduction.

## Discussion

We investigated the neural basis of time reproduction and analyzed neural correlates from rodent mPFC in a novel interval timing task. We showed that Mongolian gerbils (*M. unguiculatus*) are able to measure and reproduce durations lasting several seconds. To allow the gerbils to respond in a natural way, we used walking as a response (*Meijer and Robbers, 2014*). The task was implemented in a rodent virtual reality system (*Thurley and Ayaz, 2017*), which (1) gave us the use of a treadmill, (2) prevented landmark-based strategies for task-solving and (3) decoupled time from distance, such that the task could not be solved by path integration.

The rodents' behavior exhibited typical characteristics of time reproduction and magnitude estimation, including the regression effect, that is, the overestimation of small and underestimation of large stimuli (also known as regression to the mean, central tendency, or Vierordt's law; *von Vierordt, 1868*; *Hollingworth, 1910*; *Shi et al., 2013*; *Petzschner et al., 2015*). Neural activity in gerbil mPFC correlated with and likely contributed to time reproduction behavior. Prefrontal neurons displayed various firing characteristics, which could be grouped into a number of representative categories. Single-cell firing patterns differed between measurement and reproduction. For those cells that participated in both task phases, activity profiles never matched between task phases – although they sometimes correlated; see, for example, the ramp-type neurons in *Figure 2—figure supplement 2A and D* and *Figure 2—figure supplement 4D*. Moreover, changes in response characteristics between task phases were not coordinated across neurons, leading to low population vector correlations and no specific transition patterns between response types. Linear decomposition of the population activity, however, revealed state-space trajectories with common features in measurement and reproduction. This indicates that – despite the response heterogeneity within and between cells – the prefrontal population similarly encoded time in both task phases. Such effects on low-dimensional population activity in connection to changes of behavioral and cognitive state have been described for attention and task engagement (*Engel and Steinmetz, 2019*). Nevertheless, task-related activity in the reproduction phase was more robust and less noisy compared to the measurement phase. This was obvious from most of our analyses and could mean that the mPFC is less involved or not involved at all during the measurement phase of our task. An issue that may be resolved in further experiments.

Neural correlates of interval timing in the range of seconds have been found in several brain areas (*Merchant et al., 2013*; *Paton and Buonomano, 2018*; *Issa et al., 2020*), including prefrontal cortex (*Genovesio et al., 2006*; *Kim et al., 2013*; *Xu et al., 2014*; *Emmons et al., 2017*; *Tiganj et al., 2017*), pre-/supplementary motor cortex (*Mita et al., 2009*; *Merchant et al., 2011*), hippocampus (*MacDonald et al., 2011*), entorhinal cortex (*Heys and Dombeck, 2018*), and striatum (*Gouvêa et al., 2015*; *Mello et al., 2015*; *Bakhurin et al., 2017*; *Emmons et al., 2017*). What distinguishes our experiments from previous studies is twofold: (1) we tested time intervals on a continuous range and (2) we combined timing of an external event (measurement phase) and timing own behavior (reproduction phase), linking sensory and motor timing (*Paton and Buonomano, 2018*). Our study is therefore conceptually different from the fixed interval or discrimination tasks that have been used in most of the above timing studies. Some studies with monkeys used tasks comparable to ours but focused on intervals lasting only hundreds of milliseconds (*Jazayeri and Shadlen, 2015*; *Sohn et al., 2019*). The neural activity in primate parietal and frontal cortices observed in these studies is surprisingly similar to what we found in rodent mPFC. This is especially interesting since we tested timing of several seconds and neural dynamics typically act on much shorter time scales.

The mPFC responses we recorded during the reproduction phase are reminiscent of the neural correlates of self-initiated behavior found in rat secondary motor cortex by *Murakami et al., 2014*; *Murakami et al., 2017*. The reproduction phase in our task also involves self-initiated behavior (i.e., to stop walking). Murakami et al. found ramp-to-threshold cells similar to the one in *Figure 2C*. However, in contrast to our findings, they reported the absence of such responses in mPFC (*Murakami et al., 2017*). This discrepancy may be due to the different tasks involved: waiting for a signal to appear after a random interval in their experiments vs. responding after a previously measured interval in our case. Ramp-to-threshold responses have also been found in monkey motor cortices during various timing behaviors (*Merchant and Georgopoulos, 2006*; *Mita et al., 2009*; *Merchant et al., 2011*) and as a population pattern in lateral intraparietal cortex (*Jazayeri and Shadlen, 2015*) during time (re-)production, demonstrating their ubiquitous presence in self-initiated behaviors. Note that we also recorded negative ramp-to-threshold, that is, ramp-down, responses (*Figure 2A*). In addition, we

observed linear increasing neurons (neurons that ramp to stimulus duration-dependent levels at the same slope, e.g., *Figure 2B*) that may serve as integrators of time information provided by sequentially activated, phasically responding cells (*Figure 2—figure supplement 3*). Again, both types of neurons have been reported in other timing tasks (*Merchant et al., 2011*; *Kim et al., 2013*; *Gouvêa et al., 2015*; *Genovesio et al., 2016*). Responses with a linear ramping component (comprising both linear increasing and ramp-to-threshold) have been shown to be important for precise time decoding and to underlie low-dimensional population dynamics (*Cueva et al., 2020*).

Our time decoding analysis revealed that ramp-to-threshold and linear increasing neurons cannot by themselves explain the regression effect; rather, the combination of both response types is necessary in either a single neuron or mixed across a population of neurons. Reading out this activity will show the regression effect. We used a linear regression decoder, which takes the perspective of a neuron that forms a weighted sum of its inputs and provides a continuous time readout. This choice was motivated to gain insight into the potential mechanism of the regression effect. Despite its simplicity, it revealed that the regression effect may be the result of decoding from mixed responses. Other decoding approaches are far more efficient and precise in reading out elapsed time from neural activity like, for example, classifiers (*Bakhurin et al., 2017*; *Merchant and Averbeck, 2017*). But due to their efficiency they do not show the regression effect and would contribute less to its understanding in the present framework.

Although we cannot tell from our data how mixing is accomplished, mixed response types are compatible with theoretical models of interval timing (*Simen et al., 2011*; *Thurley, 2016*). Similarly, distributing relative and absolute timing cells across a neuronal population may yield the regression effect. However, since ramp-like responses were prominent in our data set, we consider their contribution more likely. Mixed response types are in general important in cognitive tasks (*Rigotti et al., 2013*) but have also been reported during spontaneous behavior (*Stringer et al., 2019*). Coding of variables related to cognitive functions like choice and task engagement is often distributed across brain regions (*Steinmetz et al., 2019*). Although we only recorded in one brain region and cannot comment on the distribution across the brain, our findings suggest that a local distribution of response types may also underlie cognitive functions.

Temporal scaling appears to be a general feature of timed computations in the brain as it has been described in various brain regions (*Xu et al., 2014*; *Mello et al., 2015*; *Wang et al., 2018*). It has also been demonstrated to be important for time coding in neural network models (*Bi and Zhou, 2020*). In our experiments, neuronal activity scaled and changed speed in relation to the stimulus duration in the reproduction phase only. Here, animals had to actively generate timed behavior (motor timing); in contrast to the measurement phase (sensory timing), where due to the randomization stimulus duration could not be determined in advance.

Temporal scaling and speed dependence of neural dynamics corresponded to the regression effect we observed in the behavioral data. Bayesian models (*Jazayeri and Shadlen, 2010*; *Petzschner et al., 2015*) as well as other approaches (*Bausenhart et al., 2014*; *Thurley, 2016*) have demonstrated that the regression effect may be a strategy to minimize behavioral errors. Bayesian models fuse probability distributions of the current stimulus estimate and prior knowledge. The neural representations of these probability distributions and the mechanism underlying the probabilistic computations have yet to be determined. An interesting solution was proposed by *Sohn et al., 2019* based on recordings from monkey frontal cortex. While the animals measured time intervals in a task very similar to ours, frontal cortex activity followed low-dimensional curved state-space trajectories. These curved trajectories can be interpreted as a compressed nonlinear representation of time, which when read-out appropriately during reproduction can explain regression effects seen in behavior. Our demixed PCs also showed curved trajectories during measurement (*Figure 5A*); however, trajectories from conventional PCA did not comprise such curvatures (*Figure 5D*, *Figure 5—figure supplement 5A*). These results do not necessarily contrast with those of *Sohn et al., 2019*. It is possible that – at least in our behavioral task – mPFC is not involved in the compression of the time estimate in the measurement phase. But it may contribute to reproduce the compressed estimate, for example, to signal when to stop running in the reproduction phase. The fact that we observed curved state-space trajectories in the reproduction phase suggests this possibility. At the single-neuron-level, curved state-space trajectories are supported by neurons with a ramping component. These contribute to the monotonous shape of the strongest PC (PC 1, *Figure 5B*). In combination with the non-monotonous PC 2, curved

but non-entangled state-space trajectories result that can support the mechanism suggested by *Sohn et al., 2019*.

Our results further suggest that two different types of ramping underlie the regression effect: ramping with a constant slope and ramping with a slope that depends on stimulus duration. When both response types are implemented in different neurons and distributed across the population, the amount of noise in the system determines the strength of the regression effect. Without noise, duration encoding is dominated by neurons that ramp with a constant slope and no regression effect emerges. If activity is noisy, neurons with stimulus-dependent slope contribute and the regression effect appears. Thus, the amount of noise (uncertainty about the current stimulus) determines the impact of either response type and thus the balance between current stimulus estimate and prior knowledge.

State-space trajectories for different stimulus durations were well separated during measurement (*Figure 5A*). Since stimulus duration is unknown at the beginning of the measurement phase, such duration-dependent trajectories are likely due to prior expectations about the stimulus duration. Small stimuli are typically followed by larger ones and vice versa, which may bias neural responses and behavioral estimates accordingly. Such sequential effects are known in magnitude estimation (*Bausenhart et al., 2014*; *Petzschner et al., 2015*; *Thurley, 2016*). Interestingly, when we only included ramping cells for time decoding a regression effect was also seen during measurement (*Figure 7—figure supplement 3*), implying that previous stimuli affect the current measurement. Influences of prior expectations on neural activity during time interval estimation have been reported recently (*Meirhaeghe et al., 2021*). During reproduction, trajectories were also ordered by stimulus (*Figure 5B*, *Figure 5—figure supplement 5B*), which is considered an indication that cortical dynamics are adjusted for (re-)producing different time intervals (*Remington et al., 2018*).

The present work provides insight into the neural substrate of time reproduction, including the regression effect and error minimization, in rodents. A thorough characterization of mPFC responses allowed us to show that only mixed responses in either single cells or distributed across a local population of neurons can explain the regression effect. By adjusting the relative fractions of response types, one can parameterize the strength of the regression effect and thus the fusion of stimulus estimate and prior knowledge. To resolve the specifics of the underlying neural computations will be an important direction for future research.

## Materials and methods

### Animals

The experiments in this study were conducted with three female adult Mongolian gerbils (*M. unguiculatus*) from a wild-type colony at the local animal house (referred to by IDs 10526, 11769, and 11770 throughout the article). Training started at an age of at least 4 months. The gerbils were housed individually on a 12 hr light/dark cycle, and all behavioral training and recording sessions were performed in the light phase of the cycle. The animals received a diet maintaining them at about 85–95% of their free feeding weight. All experiments were approved according to national and European guidelines on animal welfare (Reg. von Oberbayern, District Government of Upper Bavaria; reference numbers: AZ 55.2-1-54-2532-10-11 and AZ 55.2-1-54-2532-70-2016).

### Behavioral experiments

#### Experimental apparatus

Experiments were done on a virtual reality (VR) setup for rodents (*Figure 1A*). For a detailed description, see *Thurley et al., 2014*. In brief, the setup consists of an air-suspended styrofoam sphere that acts as a treadmill. On top of the sphere, the rodent is fixated with a harness that leaves head and legs freely movable. Rotations of the sphere are induced when the animal moves its legs. The rotations are detected by infrared sensors and fed into a computer to generate and update a visual virtual scene. The scene is displayed via a projector onto a projection screen that surrounds the treadmill. We used Vizard Virtual Reality Toolkit (v5, WorldViz, https://www.worldviz.com) for real-time rendering; the virtual environment was designed with Blender (v2.49b, https://www.blender.org/). Animals were rewarded with food pellets (20 mg Purified Rodent Tablet, banana and chocolate flavor, TestDiet, Sandown Scientific, UK) that were automatically delivered and controlled by the VR software.

## Behavioral paradigm

In our interval reproduction task, a rodent had to estimate the duration of a visual stimulus and reproduce it by moving along a virtual corridor. It is thus a variant of the 'ready-set-go' timing task by *Jazayeri and Shadlen, 2010*. *Figure 1A* illustrates the procedure: each trial started with the presentation of a temporal stimulus – a black screen. Animals were trained to measure its duration and not to move during this phase of the task. Stimuli were randomly chosen between 3 and 7.5 s (i.e., either 3, 3.75, 4.5, 5.25, 6, 6.75, or 7.5 s). Afterward, the visual scene switched, a virtual corridor appeared, and the animal had to reproduce the stimulus by moving through the corridor for the same duration. The animal decided on its own when to start reproducing the interval as well as when to stop. These 'reaction times' typically took a few seconds and correlated only weakly with the stimulus or the reproduced stimulus in some sessions (*Figure 1—figure supplement 2C*). If the animal continuously moved the treadmill for at least 1 s, the start of this movement was counted as the begin of reproduction. To finish reproduction, the animal had to stop for more than 0.5 s. These 0.5 s were not included in the reproduced interval. With this procedure, we avoided counting brief movements and stops as responses. *Figure 1—figure supplement 1* shows movement data from one example session.

We gave feedback to our gerbils on their reproduction performance. Following the reproduction phase, the entire projection screen was either set to green (positive, 'in') or white (negative, 'out') for 3–4 s. In addition, the animal was rewarded with one food pellet. For a reward, the reproduction had to be sufficiently close to the stimulus interval, that is, $(1 \pm k) \times$ stimulus. The width of this feedback range depended on the stimulus interval since errors increase with interval length, that is, scalar variability (*Jazayeri and Shadlen, 2010*; *Sohn et al., 2019*). Across the session, tolerance $k$ was reduced by –3% when a reward was given and extended by +3% otherwise (*Figure 1B*).

In the first trial of a session, $k$ was always set to the value from the last trial in the previous session. Adapting $k$ over a session, animals reached values of 15% and below on average (*Figure 1E*). Reward rates lay roughly between 50% and 75% (*Figure 1—figure supplement 2B*), indicating that the adaptive feedback range was successful in preventing alternative strategies such as learning the lower border of the feedback range.

The virtual corridor was designed to exclude landmark-based strategies. It was infinite and had a width of 0.5 m. The walls of 0.5 m height were covered with a repetitive pattern of black and white stripes, each with a height to width ratio of 1:5. The floor was homogeneously colored in medium light-blue and the sky was black.

By randomly changing the gain between an animals' own movement (i.e., movement on the treadmill) and movement in VR, we decorrelated movement time from virtual distance and thus prevented path integration strategies for task solving. Gain values were uniformly sampled between 0.25 and 2.25. Distributions of virtual speed, running speed, as well as their correlations with stimulus interval, reproduced duration and the bias (i.e., reproduction – stimulus) can be found in *Figure 1—figure supplement 3*. Running speed was (mostly negatively) correlated in about 25% of the sessions to stimulus and reproduction.

## Behavioral training and testing

Naive gerbils were accustomed to the VR setup in a virtual linear maze for 5–10 sessions (~2 weeks, *Thurley et al., 2014*). Then, we exposed the animals to the timing task. As a first step, we presented only stimuli of 3 and 6 s, which were easy to distinguish for the animals. The animals had to learn to either walk for a short or a long duration. Feedback was initially given with a tolerance of $k = 50\%$ and training proceeded until values below 30% were reached for at least three subsequent sessions. This training phase took about 1.5 months (ca. 30 sessions). In the second part of the training, we presented the full stimulus range for about seven sessions (1.5 weeks) to introduce the animals to stimuli on a continuous scale. Training state was quantified by the slope and CV (see also *Figure 1— figure supplement 2D*). Afterward, we implanted tetrodes into the animals' mPFC and continued with the test phase.

## Analysis of behavioral data

To compare behavioral performance across sessions and animals, we calculated different measures. To quantify the strength of the regression effect, we determined the slope of the linear regression between stimuli $s$ and their reproductions $r$. A slope of 1 would correspond to no regression

and smaller slopes to stronger regression. Variability is measured by the CV, which we calculated as $\text{CV}(r) = \text{E}_s\left[\frac{\text{STD}_s(r)}{\text{E}_r[r|s]}\right]$. Here $\text{E}_r[r \mid s]$ is the average response to a stimulus $s$ and $\text{STD}_s(r)$ the corresponding standard deviation. The ratio of both values is averaged over all stimuli, denoted by $\text{E}_s[\cdot]$. To quantify general under- or overestimation, we use $\text{BIAS}(r) = \text{E}_s\left[\text{E}_r[r \mid s] - s\right]$.

## Electrophysiological recordings

### Electrode implantation

We chronically implanted gerbils with eight tetrodes mounted to a microdrive that allowed for movement of all tetrodes together (Axona Ltd., St. Albans, UK). Tetrodes were made of 17 μm platinum-iridium wires (California Fine Wire Co.). For surgery, we anesthetized an animal with an initial dose of medetomidine-midazolam-fentanyl (0.15 mg/kg, 7.5 mg/kg, 0.03 mg/kg, s.c.) and later maintained anesthesia by 2/3 doses every 2 hr. The animal was placed on a heating pad to keep body temperature at 37°C and fixated in a stereotactic unit (Stoelting Co.). After giving local analgesia of the skull with lidocaine (Xylocain, Astra Zeneca GmbH), we drilled a hole into the skull above the right mPFC and placed tetrodes at an initial depth of 700 μm into the cortex (2.1 mm AP, –0.7 mm ML, –0.7 mm DV; *Radtke-Schuller et al., 2016*). To protect the exposed part of the brain, we used alginate (0.5% sodium alginate and 10% calcium chloride, Sigma-Aldrich) and paraffin wax. Further holes were drilled into the frontal, parietal, and occipital bone to place small jewellers' screws to help anchoring the microdrive to the skull with dental acrylic (iBond Etch, Heraeus Kulzer GmbH, Germany; Simplex Rapid, Kemdent, UK). One of the screws served as electrical ground. At the end of the surgery, anesthesia was antagonized with atipamezole-flumazenil-naloxone (0.4 mg/kg, 0.4 mg/kg, 0.5 mg/kg, s.c.). During surgery and for three postsurgical days, we gave meloxicam as a painkiller (0.2 mg/kg, s.c.). In addition, enrofloxacin antibiosis (Baytril, 10 mg/kg, s.c.) was done for 5–7 postsurgical days. The animals were allowed to recover for at least 3 days after surgery before recordings started.

### Recording procedures

Extracellular action potentials of single units were recorded at a rate of 32 kHz (Digital Lynx SX, Neuralynx, Inc). Unit activity was band-pass filtered at 600 Hz to 6 kHz. Each tetrode could be recorded differentially, being referenced by one electrode of another tetrode or the ground connected to one of the jewellers' screws. Recordings were done with Neuralynx' data acquisition software Cheetah v5.6.3 (https://neuralynx.com).

To sample different neurons throughout the experimental period, we lowered the position of the tetrodes along the dorsoventral axis of the mPFC. Lowering was done for 50 μm at the end of every second experimental session to allow for stabilization until the next experiment.

### Reconstruction of tetrode placement

Tetrode placement was verified histologically postmortem. Animals received an overdose of sodium pentobarbital and were perfused intracardially with 4% paraformaldehyde. Brains were extracted and incubated in paraformaldehyde for 1–2 days. Afterward, the brain was washed in 0.02 M phosphate buffered saline and coronal slices of 60–80 μm thickness were obtained and stained either with Neutralred or DiI (D282), NeuroTrace 500/525 Green Fluorescent Nissl Stain, and DAPI – all stains from Thermo Fisher Scientific. Histology of all animals can be found in *Figure 2—figure supplement 1A*.

## Analysis of electrophysiological data

A total of 1766 mPFC neurons were recorded over 101 experimental sessions, each with on average more than 50 trials (*Figure 1—figure supplement 2A*); animal 10526: 348 cells in 38 sessions; animal 11769: 677 cells in 32 sessions; and animal 11770: 741 cells in 31 sessions.

### Spike sorting

Spike sorting was done offline in two steps. First, data was automatically clustered with KlustaKwik (v1.6). Afterward, clusters were improved manually in 2D projections of waveform features including peak and valley, the difference between both, and energy, that is, integral of the absolute value of

the waveform, with MClust v4.3 (http://redishlab.neuroscience.umn.edu/MClust/MClust.html) under MATLAB2015b (The MathWorks, Inc). See *Figure 2—figure supplement 1* for example spike clusters.

Quality of spike sorting was assessed by calculating (1) rate of interspike interval (ISI) violations (spikes with ISI < 1.5 ms are assumed to come from different neurons) and (2) calculating the fraction of spikes missing assuming a symmetric distribution of amplitudes (for details, see *Hill et al., 2011*). We excluded clusters with ISI violations > 0.5 and amplitude cutoff > 0.1. Both measures were calculated using the implementation in the Allen Institute ecephys spike sorting Python modules (https://github.com/AllenInstitute/ecephys_spike_sorting; *Siegle et al., 2021*). Also, only units with stable firing throughout a session entered further analysis. A unit was considered stable if spike counts in 1 min windows did not drop below 4 standard deviations from the mean session firing rate.

## Spike density functions

We determined spike density functions (SDFs) for each task phase separately. To calculate an SDF, spikes were either aligned at the begin or the end of the task phase. Then, spikes were counted in 100 ms windows for all trials at the same stimulus and divided by window width to gain firing rates. The windows were right aligned (looking into the past) to gain causal SDFs. To avoid edge effects due to response variability at the same stimulus, trials were scaled to the average response for the reproduction phase. During the measurement phase, trials had the same duration by design. For visualization only (*Figure 2*, *Figure 2—figure supplement 2*-*Figure 2—figure supplement 4*), SDFs were smoothed with a half Gaussian kernel of 3-bin standard deviation whose direction matched the right alignment of the window for spike counting, that is, which was looking into the past.

For the analyses in Figures 3–7 and accompanying supplementary figures, SDFs were z-scored to account for cell-specific differences in firing rate. In addition, we resampled to same number of bins (time-normalized) the SDFs of different cells and for different stimuli to be able to compare data from stimuli and responses of different duration.

For the population plots in *Figure 3A–C*, neurons were sorted by the angle between their demixed PCA scores for the first two time course-related PCs (see below for the description of the demixed PCA). This takes into account the full response profile instead of single features like the peak firing rate.

Control SDFs used in *Figure 4* and *Figure 7—figure supplement 1* were generated by (1) shuffling SDFs for each stimulus across cells ('shuffled data') and (2) shuffling single responses over time ('noise').

## Single-cell responses to running speed

To determine the influence of running speed on the firing rate of a neuron (speed response function), we counted the number of spikes that occurred at a certain running speed (5 cm/s bins, minimum speed 10 cm/s, maximum speed 1 m/s) and divided the count by the total duration the animal moved at this speed. This response function was hence independent of trial and stimulus interval. To assess statistical significance of speed modulation, we (1) shuffled spikes times, (2) recalculated the firing rate as a function of speed in each bin, and (3) determined the variance in firing rate of the shuffled speed response function (*Saleem et al., 2013*). To have a robust shuffle control value, we repeated this procedure 10 times and used the shuffled speed response function with the largest variance. Using Levene's test, this variance was compared to that of the true speed response function. We tested for variances larger than the shuffle control only. A speed modulation index in firing rate was calculated from the average firing rate in the 0–10 and 90–100 percentiles of the speed response function as $(r_{90} - r_{10})/(r_{90} + r_{10})$.

## Single-cell and population correlations

In *Figure 3*, we report different Pearson correlations for the single-cell and population data. All correlations were calculated on the time-normalized SDFs. Pairwise correlations were determined for the responses to all stimuli between all pairs of neurons (insets in *Figure 3A–C*). The population vector correlation in *Figure 3D* was calculated between the activity of all neurons in one time bin during measurement and the corresponding time bin during reproduction, that is, correlating columns in *Figure 3A and C*. In *Figure 3E*, single-cell activity across all stimuli was correlated between measurement and reproduction, that is, correlating rows in *Figure 3A and C*. In *Figure 3F*, population vector

correlations were calculated for all stimulus pairs, that is, correlating all points belonging to a particular stimulus in *Figure 3A* or B to those belonging to another stimulus. Similarly, correlations for all pairs of stimuli were determined for each individual cell in *Figure 3G*.

## Principal component analysis

To gain a reduced representation of the collective population activity over time, we applied demixed and conventional PCA in the $T$-dimensional space where each dimension represents the firing rate at a different time point. A neuron's activity pattern is hence represented as one single point in this space and a PC will be a component that has $T$ points and evolves over time. The first PC will be the temporal pattern of activity that explains most variance; the second PC the one orthogonal to the first with second most variance and so on. By this method, PCs represent collective population activity over time. With demixed PCA, this population activity can be separated into components related to the stimulus interval and those related to the overall time course of the population activity independent of the stimulus.

Demixed PCA was performed separately for measurement and reproduction on the SDFs of all recorded neurons aligned at the respective onsets (*Figure 5*); see *Kobak et al., 2016* for a detailed description. We used the demixed PCA implementation available at https://github.com/machenslab/dPCA (*Kobak et al., 2021*). When we applied demixed PCA on data from individual animals, results were similar (*Figure 5—figure supplement 1 Figure 5—figure supplement 2*, and *Figure 5—figure supplement 3*). Conventional PCA was also done separately for measurement and reproduction and on the SDFs of all recorded neurons aligned at the respective onsets (*Figure 5—figure supplement 5*).

Bootstrapping was done by performing demixed PCA on 1000 random subsets comprising each 10% of the whole data set. Subsets were picked in a stratified way, that is, accounting for the different numbers of cells recorded in each animal. The function StratifiedShuffleSplit from scikit-learn was used for picking the subsets. Results were similar for 5% and 20% subsets.

## Tensor maximum entropy surrogates

To test whether population responses contained collective contributions beyond what is expected from pooling single neurons, we generated control surrogate data according to *Elsayed and Cunningham, 2017a*. Random tensor maximum entropy surrogate samples were drawn that preserved the stimulus tuning of single neurons, correlations of single-cell firing rates across time and signal correlations across neurons. The implementation we used is available at https://github.com/gamaleldin/rand_tensor (*Elsayed and Cunningham, 2017b*).

## Categorization of response types

We categorized cells into different response types by their score values for specific PCs: time course-related PC 1 and stimulus PC 1 for measurement and time course-related PCs 1–3 and stimulus PC 1 for reproduction. Since the scores for those PCs had single peaked distributions, displaying no obvious clusters or response groups (*Figure 5—figure supplement 4*), we used the following procedure to construct response categories: first, a cell's responses in either task phase were reconstructed as a linear combination from the abovementioned PCs weighted by the respective PCA scores. Then the variance of the cell's response was determined, which was explained by this reconstruction. If this explained variance was below the cumulative overall explained variance for the PCs (measurement: 6%, reproduction: 28%; *Figure 5—figure supplement 1*), the cell was assigned to the 'unrelated activity' category (*Figure 5—figure supplement 4*). Otherwise, first the strongest PC (the one with the largest absolute scores) was found and then the angles between this PC and each of the other PCs were determined (calculated on the absolute values). If any of those angles was above 22.5°, the other component was counted as contributing (*Figure 5—figure supplement 4C*). In total, $2^2 = 4$ different response types were possible in the measurement phase and $2^4 = 16$ in the reproduction phase, that is, categories ranging from 'unrelated activity' with no overlap with any of the PCs to activity explained by all PCs used for categorization.

Categories were validated by categorizing every cell by its scores for each of the 1000 bootstrapped demixed PCAs described above. Finally, the category with maximum likelihood was assigned to the cell.

In addition, we compared the number of cells in each category to a random prediction. For that, we constructed random surrogate data by shuffling SDFs across stimuli and cells, performed demixed PCA on this data, categorized each 'surrogate cell,' and counted the number of cells in each category. From 1000 such shufflings, we got distributions of by chance expected cell counts in each category, which we used to determine p-values for the count in the original data. A level of 5% was chosen and indicated as significant in *Figure 6*.

## Time decoding

To decode elapsed time, we used multiple linear regression (Wiener filter; *Glaser et al., 2020*, https://github.com/kordinglab/neural_decoding) between time points and the spike responses (SDFs) of all neurons, that is, the SDFs of each neuron were weighted such that elapsed time could be most precisely decoded. The SDFs of all neurons were combined in a matrix $\mathbf{R}$ with the individual SDFs as columns, such that the matrix had as many rows as time points and as many columns as neurons. Representing ongoing time as a vector $\mathbf{t}$, the regression problem reads

$$\mathbf{R} \cdot \beta = \mathbf{t}, \tag{1}$$

with $\beta$ being the vector comprising the weights for each neuron (SDF), which can be fit by least squares.

The above equation deals only with one SDF per neuron, that is, the response at one stimulus interval. However, we did not want to find individual weights for each stimulus but one weight for all stimuli per neuron. Therefore, we concatenated a neuron's SDFs for all stimuli, such that $\mathbf{R}$ had as many rows as time points × stimuli. The weights $\beta$ were then fit. This treated the whole data set as a reference (prior). During decoding ongoing time at a particular stimulus, we plugged in only the SDF at this stimulus into the left side of *Equation 1* and received a vector of decoded time points. Fitting and decoding was done on random subsets of 20 cells (1000 bootstrap runs), from which we extracted average and standard deviation. To avoid overfitting, zero-mean Gaussian noise with $\sigma = 0.5$ was added to the SDFs. Note that the SDFs were z-scored.

## Additional notes on data analysis

Data analysis was done with Python 2.7 using Matplotlib 2.2, Numpy 1.15, Pandas 0.24, Scipy 1.2, Scikit-learn 0.20, and Statsmodels 0.10 – in addition to abovementioned packages. If p-values are not provided, significance is indicated by * $p<0.05$, ** $p< 0.01$, *** $p<0.001$.

## Acknowledgements

We thank Tobias Bernklau for help with data analysis and Moritz Dittmeyer for help with the schematic drawing of the setup.

## Additional information

### Funding

| Funder | Grant reference number | Author |
|---|---|---|
| Bundesministerium für Bildung, Wissenschaft und Forschung | 01GQ1004A | Josephine Henke Kay Thurley |

The funders had no role in study design, data collection and interpretation, or the decision to submit the work for publication.

### Author contributions

Josephine Henke, Data curation, Investigation, Methodology, Writing – review and editing; Raven Bunk, Dina von Werder, Formal analysis, Software, Writing – review and editing; Stefan Häusler, Formal analysis, Methodology, Writing – review and editing; Virginia L Flanagin, Conceptualization, Methodology, Writing – review and editing; Kay Thurley, Conceptualization, Data curation, Formal

analysis, Methodology, Project administration, Software, Supervision, Writing - original draft, Writing – review and editing

### Author ORCIDs
Dina von Werder  http://orcid.org/0000-0002-4193-5203
Stefan Häusler  http://orcid.org/0000-0003-0772-8662
Virginia L Flanagin  http://orcid.org/0000-0002-6677-459X
Kay Thurley  https://orcid.org/0000-0003-4857-1083

### Ethics
All experiments were approved according to national and European guidelines on animal welfare (Reg. von Oberbayern, District Government of Upper Bavaria; reference numbers: AZ 55.2-1-54-2532-10-11 and AZ 55.2-1-54-2532-70-2016).

### Decision letter and Author response
Decision letter https://doi.org/10.7554/eLife.71612.sa1
Author response https://doi.org/10.7554/eLife.71612.sa2

---

## Additional files

### Supplementary files
Transparent reporting form

### Data availability
Raw data for this study are available at https://doi.org/10.12751/g-node.tarvrs (Henke et al., 2021). In addition, source data are given when mentioned in the figure captions.

The following dataset was generated:

| Author(s) | Year | Dataset title | Dataset URL | Database and Identifier |
|---|---|---|---|---|
| Henke J, Bunk D, von Werder D, Häusler S, Flanagin V, Thurley K | 2021 | Data for Distributed coding of duration in rodent prefrontal cortex during time reproduction | https://doi.org/10.12751/g-node.tarvrs | tarvrs, 10.12751/g-node.tarvrs |

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
