## [Editor Report]

This study investigates the neural underpinnings of the bias property of timing, namely an overestimation for short and underestimation for long intervals, during an interval reproduction task in the medial prefrontal cortex of gerbils. The key novel result is that only neural populations with mixed responses, including ramping activity with linear increasing and slope-changing modulations as a function of reproduced durations, can encode the bias effect. Overall, experiments and data analysis are technically sound, and the conclusions well supported.

---

## [Decision Letter]

**Decision letter after peer review:**

Thank you for submitting your article "Distributed coding of stimulus magnitude in rodent prefrontal cortex" for consideration by *eLife*. Your article has been reviewed by 3 peer reviewers, including Hugo Merchant as Reviewing Editor and Reviewer #1, and the evaluation has been overseen by Michael Frank as the Senior Editor.

Essential revisions:

1) The main weakness of the paper is a conceptual problem, which requires clarification. Both the title and the introduction point to magnitude quantification, which in general exhibits the regression effect (Vierordt's law). Interval timing is certainly a form of magnitude quantification and exhibits the regression effect. However, any neural correlate of the regression effect in interval timing does not necessarily generalizes to other forms of magnitude quantification. The way the findings of this paper are framed in the title, abstract, introduction, and discussion seem to make that generalization, and that claim is not supported because the data shown has a narrower scope.

2) It is necessary to clearly demonstrate that the behavioral protocol produces a robust, reliable time interval reproduction. Is also fundamental to provide a valid justification for such low levels of performance. It is especially important to clarify this point before analyzing the potentially associated neuronal activity. It is important to understand how stable the behavior is in a session-by-session base. We suggest including a full description of the learning curve. This point is especially relevant for the population analysis, where the activity of cells collected in different sessions was pooled together. In addition, it's well known that in behavioral protocols like this, behavior tend to stereotype, making it difficult to determine if the neural activity is associated with a particular behavioral variable (such as speed or distance) or elapsed time. Hence, it would be necessary to formally assess how behavioral variables covariate (or not) with time. For this, examples of single sessions are not sufficient. It is necessary to clarify several things. For example, the slopes of the linear regressions for dozens of sessions are presented (Figure 1E) but is not clear if there was a behavioral evolution over the training. For another example, animal 10526 presented high levels of variability in many variables (Figure S2), hence, it would be important to know which sessions were used for the analysis of the neural data.

3) A main conclusion of the paper is that only neural populations with mixed responses, including ramping activity with linear increasing and slope-changing modulations as a function of reproduced durations, can encode the regression effect. How the prior knowledge about the distribution of used intervals in the task is combined with the actual measurement of the passage of time? It is not evident on the last part of the paper how the slope changing cells, associated with a prior signal, are combined with the linearly increasing ramping cells to generate both the regression effect and a sufficiently accurate representation of produced duration. Sessions within the same animal with different slopes in the regression effect show different proportions of the two cell types? How is the mixing accomplished?

4) During reproduction, the percent of neurons categorized with time-dependent PC 1 and stimulus PC 1 responses is < 1%, which is one of the two combinations that gives the best regression effect in the decoding. These linearly increasing ramping cells were very common during measurement and may be encoding elapsed time since the stimulus onset. In contrast, most of the responses in reproduction were slope changing cells, reaching a peak of activation close to the end of the produced interval. This type of responses has been observed when a prediction of time to an event is needed (time-to-contact cells Merchant and Georgopoulos, 2006). A key issue, then, is whether the decoding method used to reconstruct elapsed time is capturing the prediction to an event signal as a regression effect because is aligned to the onset of the time interval, instead of to the end of the produced interval, which is probably the task event that the cells are encoding.

5) The authors say the conventional PCA did not show curved trajectories (line 418, Discussion, Pg. 17) for the measuring phase. However, that is certainly not the case for the reproduction phase, where the decoding analysis was successfully performed, and the regression effect was observed. I don't see the present results as alternative to the hypothesis by Sohn et al., 2019., but as a quite interesting step forward from that. It is possible that what the PFC is encoding in this task is an estimate of when to stop moving. In that case, the neural activity would be equivalent to that observed in Sohn et al., 2019. And perhaps warped and transformed by down-stream reading areas similarly to the "curved manifold projected onto a line". It would be very interesting if slope-changing cells were contributing to a more curved shape of the population trajectory, since they can be encoding a prior as the authors suggest. The results of the decoding analysis which uses a linear (i.e. lower dimensional) decoder indirectly suggest this.

6) Despite the clear differences in neural activity between the two phases presented in figures 2 to 4, authors named the following section "Population activity shares similar components for measurement and reproduction" (page 8 line 58). However, that entire section, that includes figures 5 and 6 and several supplementary figures (e.g. S8 and S9), shows notorious differences between the two phases. From the explained variance (Figures 5, S8, S9) to how single cell activity contributes to the population representations, there are striking differences between the two behavioral phases. Authors must clarify what do they mean by "activity shares similar components" or change the name of the section for something that better represents the data.

7) Elapsed time decoding from populations of cells using classifiers suggest that different ramping cell types can contribute to represent the passages of time and that the decoding error can explain the error in produced interval (Merchant and Averbeck, 2017; Bakhurin et al., 2017). Based on this and the previous comment I am wondering whether a classifier is not a better option for decoding than the linear decoder.

---

## [Author Response]

Essential revisions:1) The main weakness of the paper is a conceptual problem, which requires clarification. Both the title and the introduction point to magnitude quantification, which in general exhibits the regression effect (Vierordt's law). Interval timing is certainly a form of magnitude quantification and exhibits the regression effect. However, any neural correlate of the regression effect in interval timing does not necessarily generalizes to other forms of magnitude quantification. The way the findings of this paper are framed in the title, abstract, introduction, and discussion seem to make that generalization, and that claim is not supported because the data shown has a narrower scope.

We agree that our findings in particular with regard to the neural correlates are more in the scope of interval timing. Therefore, we have toned down the emphasis on magnitude quantification in connection with our findings and now take a stronger interval timing/reproduction perspective. We changed the title and edited abstract, introduction, results and discussion.

2) It is necessary to clearly demonstrate that the behavioral protocol produces a robust, reliable time interval reproduction. Is also fundamental to provide a valid justification for such low levels of performance. It is especially important to clarify this point before analyzing the potentially associated neuronal activity. It is important to understand how stable the behavior is in a session-by-session base. We suggest including a full description of the learning curve. This point is especially relevant for the population analysis, where the activity of cells collected in different sessions was pooled together.

The gerbils display very well and reliable time reproduction behavior in our experiments. Task performance of the gerbils in our behavioral paradigm is comparable to previous studies conducted with humans and non-human primates on time interval reproduction (Jazayeri and Shadlen 2010 *Nature Neuroscience,* Jazayeri and Shadlen 2015 *Current Biology* and Sohn et al., 2019 *Neuron*). This is indicated by the slope of the linear fit between stimulus interval and reproduced interval being between 0.5 and 1, and by the small coefficients of variation (CV) being below 0.2 on average. In own time reproduction experiments with humans, performance levels were similar (Thurley and Schild 2018 *Sci Rep*).

To better illustrate the behavioral performance, we now plot the development of slope and coefficient of variation across sessions for each animal in Figure 1—figure supplement 2D. In these plots we also include the final training sessions, when we started presenting the whole stimulus distribution. An ongoing improvement can be seen for each animal during training. Data at even earlier stages of training are not included in the plot, since these were not tracked systematically. Training data did not appear in the earlier version of the manuscript and only does in Figure 1—figure supplement 2D now.

The impression of “low levels of performance” most likely stems from the “percentage of hits” in each session that we presented in former Figure S2B. Hits in our paradigm can not be interpreted as for binary decision tasks, this means that 50% performance does not correspond to chance performance.

We used an adaptive feedback protocol to decide whether a reproduced interval is close enough to the stimulus interval and hence rewarded or not. Rewarding is necessary to keep the animals motivated. However, if rewards would always be given at a fixed distance off the stimulus, e.g. 15%, the animals could learn to reproduce 15% shorter intervals than the stimulus interval. Adaptive feedback avoids this situation as the distance from stimulus that needs to be reached to receive a reward changes with every (non-)rewarded trial. Effectively, rewards will be given in only a bit more than 50% of the trials. See the following papers that used a similar feedback protocol:

Jazayeri and Shadlen 2010 *Nature Neuroscience*: “As such, every subject’s performance for every session yielded approximately 50% positively reinforced trials.”

Jazayeri and Shadlen 2015 *Current Biology*: “The scaling constant was adjusted so that approximately 60% of trials were rewarded.”

We understand that the term “hits” is misleading, so we now use “rewards” instead (Figure 1—figure supplement 2B).

In addition, it's well known that in behavioral protocols like this, behavior tend to stereotype, making it difficult to determine if the neural activity is associated with a particular behavioral variable (such as speed or distance) or elapsed time. Hence, it would be necessary to formally assess how behavioral variables covariate (or not) with time. For this, examples of single sessions are not sufficient. It is necessary to clarify several things. For example, the slopes of the linear regressions for dozens of sessions are presented (Figure 1E) but is not clear if there was a behavioral evolution over the training. For another example, animal 10526 presented high levels of variability in many variables (Figure S2), hence, it would be important to know which sessions were used for the analysis of the neural data.

All behavioral data reported is from the sessions in which also recordings were performed, except for the new Figure 1—figure supplement 2D in which we now also present behavioral data from the final training sessions. No data from training sessions is used in the paper at any other point.

We always excluded sessions with too low performance from the data set. In the revision, we reconsidered behavioral performance and decided to exclude another 4 border-line sessions. Our data set now includes 1766 cells recorded in 101 sessions.

We randomly changed the gain between virtual movement and treadmill movement in every trial. This efficiently decorrelates time and distance (stimulus interval and path length, Figure 1—figure supplement 1C, formerly Figure S1C). This is a typical procedure to enforce time reproduction and avoid e.g. distance reproduction as an alternative strategy for task solving in virtual reality experiments (e.g. Petzschner and Glasauer 2011 *J Neurosci*, Thurley and Schild 2018 *Sci Rep*, Robinson and Wiener 2021 *NeuroImage*). Since neural responses may nevertheless be related to, running speed, we extended the analysis in this respect.

3) A main conclusion of the paper is that only neural populations with mixed responses, including ramping activity with linear increasing and slope-changing modulations as a function of reproduced durations, can encode the regression effect. How the prior knowledge about the distribution of used intervals in the task is combined with the actual measurement of the passage of time? It is not evident on the last part of the paper how the slope changing cells, associated with a prior signal, are combined with the linearly increasing ramping cells to generate both the regression effect and a sufficiently accurate representation of produced duration. Sessions within the same animal with different slopes in the regression effect show different proportions of the two cell types? How is the mixing accomplished?

This is a very interesting question! We do not have a specific mechanistic hypothesis as to how the mixing occurs, but we also do not think we can tell from our data. Our data and analyses in Figure 7F only indicates that mixing linear increasing and slope-changing cells at different fractions allows for parameterizing the strength of the regression effect. How mixing is accomplished, we can not directly tell from the present experiments, but we want to explore this question in future work.

We are also hesitant to draw direct conclusions from our data about the relationship between the proportions of active cell types and the regression effect in a given session. Our recording technique only allowed the recording of different cells in each session due to changes of electrode position between sessions. Cell numbers and types most likely depend on the specific subset of cells we were able to recorded rather than on a potentially dynamic recruitment of cells in each session.

A conclusive experiment would need to systematically record from the same cells across several sessions with changing strength of the regression effect.

We now explicitly point out in the Discussion that “we can not tell from our data how mixing is accomplished”.

4) During reproduction, the percent of neurons categorized with time-dependent PC 1 and stimulus PC 1 responses is < 1%, which is one of the two combinations that gives the best regression effect in the decoding. These linearly increasing ramping cells were very common during measurement and may be encoding elapsed time since the stimulus onset. In contrast, most of the responses in reproduction were slope changing cells, reaching a peak of activation close to the end of the produced interval. This type of responses has been observed when a prediction of time to an event is needed (time-to-contact cells Merchant and Georgopoulos, 2006). A key issue, then, is whether the decoding method used to reconstruct elapsed time is capturing the prediction to an event signal as a regression effect because is aligned to the onset of the time interval, instead of to the end of the produced interval, which is probably the task event that the cells are encoding.

Indeed, decoding from slope-changing cells allows to predict the final time point in the reproduction interval (time to event, time to contact) precisely. We tested this training a classifier to differentiate between the final time bin of reproduction versus all other time bins (25 bins in total) for data aligned at the end of the time interval. This shows that the slope-changing cells contain information about the “contact” to the final time point. However, this does not translate into the (behavioral) regression effect if one tries to decode elapsed time from the same cells. As we show in Figure 7C, the readout from slope-changing cells will be the same final time point for all stimulus durations, i.e. regression slope = 0. The insight we gained from our decoding analyses is that in order to explain the milder regression slopes observed behaviorally, slope-changing and linear-increasing cells need to be readout together. We emphasize this in the Discussion now.

5) The authors say the conventional PCA did not show curved trajectories (line 418, Discussion, Pg. 17) for the measuring phase. However, that is certainly not the case for the reproduction phase, where the decoding analysis was successfully performed, and the regression effect was observed. I don't see the present results as alternative to the hypothesis by Sohn et al., 2019., but as a quite interesting step forward from that. It is possible that what the PFC is encoding in this task is an estimate of when to stop moving. In that case, the neural activity would be equivalent to that observed in Sohn et al., 2019. And perhaps warped and transformed by down-stream reading areas similarly to the "curved manifold projected onto a line". It would be very interesting if slope-changing cells were contributing to a more curved shape of the population trajectory, since they can be encoding a prior as the authors suggest. The results of the decoding analysis which uses a linear (i.e. lower dimensional) decoder indirectly suggest this.

We also performed demixed and conventional PCA only with specific response types. Since we determined response types from the principal components in the whole data set, restriction to or exclusion from certain response types means focusing or removing the related principal components.

The slope-changing cells contribute mostly a monotonous change (PC 1 in Figure 5B). This monotonous change is not completely linear and hence contributes to a curved shape of the population trajectory. The most important contribution to the trajectory, however, is the monotonous change. In combination with the non-monotonous PC 2 it results in a smooth and curved population trajectory. If we remove all cells with a slope-changing component, i.e. all response types with a contribution of PC 1, PC 2 and 3 become dominant. Since both PC 2 and PC 3 are non-monotonous they form a tangled population trajectory. From that we conclude that a slope-changing component is essential for a smooth, untangled but curved population trajectory that could support the regression effect as suggested by Sohn et al., 2019.

We added the above insights to the discussion. We also agree with the reviewers that our results do not represent an alternative to Sohn et al., 2019 and have therefore refined and tempered the discussion in this regard.

6) Despite the clear differences in neural activity between the two phases presented in figures 2 to 4, authors named the following section "Population activity shares similar components for measurement and reproduction" (page 8 line 58). However, that entire section, that includes figures 5 and 6 and several supplementary figures (e.g. S8 and S9), shows notorious differences between the two phases. From the explained variance (Figures 5, S8, S9) to how single cell activity contributes to the population representations, there are striking differences between the two behavioral phases. Authors must clarify what do they mean by "activity shares similar components" or change the name of the section for something that better represents the data.

We agree that the subsection’s title was not appropriate and changed it to “Collective population activity in the different task phases”. We also changed the descriptions in the paragraph to more strongly emphasize the similarities and differences in collective activity between the two task phases.

7) Elapsed time decoding from populations of cells using classifiers suggest that different ramping cell types can contribute to represent the passages of time and that the decoding error can explain the error in produced interval (Merchant and Averbeck, 2017; Bakhurin et al., 2017). Based on this and the previous comment I am wondering whether a classifier is not a better option for decoding than the linear decoder.

We performed decoding analyses using a support-vector machine classifier (Bakhurin et al., 2017) implemented in Glaser et al., 2020, https://github.com/kordinglab/neural_decoding. Indeed, with such a classifier one could decode elapsed time much more veridical than with a linear regression decoder. We show in Author response image 1, a variant of Figure 7 from the main text in which we (1) split 7.5 s of real time into 25 bins and (2) decoded time with a support-vector machine classifier.

**Author response image 1. sa2fig1:** 

The readout was similar whether decoding was done from all data (A), linear increasing cells (B), or slope-changing cells (C). Final values of decoded time displayed large slopes indicating veridical values close to real time (D). For slope-changing cells decoding deviated markedly from theoretical prediction (C). This is likely due to the classifier exploiting subtle differences (partly random and due to noise) between recorded cells for decoding time. This also explains the overall better decoding performance of a classifier compared to a linear regression decoder. Despite its lower performance, in our view, the insight gained from a linear regression decoder with respect to the nature of the regression effect is larger than from a classifier. Also the mechanistic interpretation is different. A linear regression decoder may be implemented by a readout unit receiving continuous inputs and decoding elapsed time. A classifier would correspond to several readout units decoding a different time point each.

We extended the Discussion by this aspect but we did not include the figure above into the manuscript.